

# Confinement and false vacuum decay on the Potts quantum spin chain

Octavio Pomponio[1*], Anna Krasznai[2,3†] and Gábor Takács[2,3,4‡]

**1** SISSA and INFN, Sezione di Trieste, via Bonomea 265, I-34136, Trieste, Italy
**2** Department of Theoretical Physics, Institute of Physics,
Budapest University of Technology and Economics,
Műegyetem rkp. 3., H-1111 Budapest, Hungary
**3** BME-MTA Statistical Field Theory 'Lendület' Research Group,
Budapest University of Technology and Economics,
Műegyetem rkp. 3., H-1111 Budapest, Hungary
**4** MTA-BME Quantum Dynamics and Correlations Research Group,
Budapest University of Technology and Economics,
Műegyetem rkp. 3., H-1111 Budapest, Hungary

★ octavio.pomponio@gmail.com , † anna.krasznai@edu.bme.hu , ‡ takacs.gabor@ttk.bme.hu

## Abstract

We consider non-equilibrium dynamics after quantum quenches in the mixed-field three-state Potts quantum chain in the ferromagnetic regime. Compared to the analogous setting for the Ising spin chain, the Potts model has a much richer phenomenology, which originates partly from baryonic excitations in the spectrum and partly from the various possible relative alignments of the initial magnetisation and the longitudinal field. We obtain the excitation spectrum by combining semiclassical approximation and exact diagonalisation, and we use the results to explain the various dynamical behaviours we observe. Besides recovering dynamical confinement, as well as Wannier-Stark localisation due to Bloch oscillations similar to the Ising chain, a novel feature is the presence of baryonic excitations in the quench spectroscopy. In addition, when the initial magnetisation and the longitudinal field are misaligned, both confinement and Bloch oscillations only result in partial localisation, with some correlations retaining an unsuppressed light-cone behaviour together with a corresponding growth of entanglement entropy.

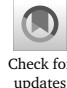

# 1 Introduction

Confinement is a central concept in the theory of strong interactions, a.k.a. quantum chromodynamics, which leads to the absence of quarks (and gluons) from the spectrum of experimentally observed particles [1]. The underlying mechanism is based on a linear potential, which can also be realised in condensed matter systems, as shown first for the scaling limit two-dimensional Ising model in [2], with a general picture in 2D quantum field theories given in [3].

Confinement also strongly affects non-equilibrium dynamics. For the ferromagnetic quantum Ising spin chain [4], it suppresses entropy growth and quasiparticle propagation. Dynamical confinement and its effects have recently been studied and confirmed in numerous systems [5–18]. Besides dynamical confinement, another effect that leads to the suppression of quasiparticle propagation is Wannier-Stark localisation, as demonstrated for the Ising spin chain in [6,10]. Wannier-Stark localisation occurs when particles in a periodic potential (e.g. in a lattice) move under the influence of a uniform external field, whereas the periodicity of the dispersion relation over the Brillouin zone leads to spatial localisation, restricting particle motion and significantly altering transport properties. In contrast to the usual confinement, which is also present in the scaling field theory limit, Wannier-Stark localisation results from Bloch oscillations specific to condensed matter systems defined on a lattice [19]. Bloch oscillations were also shown to suppress the decay of the false vacuum under the standard scenario of bubble nucleation [20,21] by preventing the expansion of the nucleated bubbles of the true vacuum [22]. Transport properties were also found to be strongly suppressed both by confinement and Wannier-Stark localisation [6,10], albeit recently it was pointed out that escaping fronts of significant magnitude appear for local quantum quenches [23].

The quantum Ising spin chain, considered in most previous studies, is a paradigmatic model of quantum many-body systems. However, its simplicity also severely restricts its dynamics; the analogy to confinement in QCD is restricted by the $\mathbb{Z}_2$ valued order parameter, which corresponds to setting the number of colours to two. As a result, the 'hadron' spectrum is restricted to mesonic excitations, and there are no analogues of the baryons. Motivated by this, here we consider the mixed-field 3-state Potts quantum spin chain, corresponding to three colours, where the 'hadron' spectrum contains both mesonic and baryonic excitations, which have already been studied in the scaling limit [24–27].

We approach non-equilibrium dynamics by considering quantum quenches, i.e., a sudden change in the Hamiltonian [28,29], which is a paradigmatic protocol that is routinely engineered in experiments on closed quantum systems [30–37]. We restrict our consideration here to the translationally invariant case, a.k.a. a global quantum quench, following the spirit of [4]. Compared to the Ising model, it turns out that the Potts chain displays a much more diversified phenomenology due to the larger number of colours.

While global quenches in the paramagnetic phase of the Potts chain have already been investigated [38,39], in this work, we study quantum quenches in the ferromagnetic phase, where the presence of both transverse and longitudinal fields is expected to result in dynamical confinement and/or Wannier-Stark localisation. However, it turns out that the phenomenology is much richer due to the possibility of misalignment between the initial magnetisation and the longitudinal field, leading to what we call 'oblique' quenches. In this case, confinement or Wannier-Stark localisation only results in partial localisation, with some correlations retaining an unsuppressed light-cone behaviour together with a corresponding growth of entanglement entropy.

The outline of the work is as follows. In Section 2, after introducing the model and reviewing its main features, we describe the quench protocols and summarise their phenomenology. We then give a semiclassical description of the excitation spectrum in Section 3, the results of

which are used in Section 4 to interpret and explain the phenomenology of global quantum quenches in the confining case of the Potts quantum spin chain. We summarise our conclusions in Section 5. To keep the main line of the argument uninterrupted, details regarding exact diagonalisation computations and the iTEBD simulations are relegated to the Appendix.

## 2 Quench phenomenology

### 2.1 The mixed-field three-state Potts quantum spin chain

The mixed-field 3-state Potts quantum spin chain is defined on the Hilbert space

$$\mathcal{H} = \bigotimes_{i=1}^{L} [\mathbb{C}^3]_i \,, \tag{1}$$

where $i$ denotes the site index along a chain of length $L$. The local Hilbert space $\mathbb{C}^3$ at each site $i$ has a basis $|\mu\rangle$ with $\mu = 1, 2, 3$ representing the spin states. The system's dynamics is governed by the Hamiltonian, assuming periodic boundary conditions $P_{L+1}^{\mu} \equiv P_1^{\mu}$,

$$H = -J \sum_{i=1}^{L} \sum_{\mu=1}^{3} (P_i^{\mu} P_{i+1}^{\mu} + h_{\mu} P_i^{\mu}) - J g \sum_{i=1}^{L} \tilde{P}_i \,, \tag{2}$$

where the traceless operators $P_i^{\mu} = |\mu\rangle_{ii}\langle\mu| - 1/3$ tend to project the spin at site $i$ along the 'direction' $\mu$. The first term in (2) promotes a ferromagnetic ground state, with all spins polarised in one of the three directions. When present, the longitudinal fields $h_{\mu}$ break the degeneracy between the three possible spin orientations. In contrast the traceless operator $\tilde{P}_i = |\mu_0\rangle_{ii}\langle\mu_0| - 1/3$ couples to a 'transverse field' $g$, and favours the direction $|\mu_0\rangle_i \equiv \sum_{\mu} |\mu\rangle_i / \sqrt{3}$. The relative strength of the first and third terms is regulated by the coupling $g$. These terms compete with each other, and in the absence of the 'longitudinal' magnetic fields $h_{\mu}$, their competition leads to a phase transition with a critical point at $g = 1$. This critical point signifies a phase transition from a paramagnetic (PM) phase for $g > 1$ to a ferromagnetic (FM) phase for $g < 1$. In the PM phase, there is a unique $\mathbb{S}_3$-symmetric vacuum, whereas, in the FM phase, there are three vacua that become degenerate in the thermodynamic limit $L \to \infty$. The order parameter for this transition is the magnetisation $M_{\mu} = \langle P_i^{\mu} \rangle$, and the quantum critical point separating these phases can be described by a conformal field theory (CFT) with a central charge $c = 4/5$. Away from the critical point $g = 1, h_{\mu} = 0$, the $q = 3$ Potts quantum spin chain is non-integrable, unlike the Ising model, where integrability is preserved unless a longitudinal field is introduced.

From now on, we set the constant $J = 1$ together with $\hbar = 1$, corresponding to a choice of energy and time units.

The spectrum of the purely transverse chain ($h_{\mu} = 0$) can be constructed perturbatively in both the extreme paramagnetic $g \gg 1$ and extreme ferromagnetic $g \ll 1$ limits [40]. In the paramagnetic phase, the ground state is non-degenerate, and the quasiparticle spectrum consists of doubly degenerate *magnons*. In contrast, in the ferromagnetic phase, there are three degenerate ground states (vacua) and the elementary excitations are *kinks* $K_{\mu\nu}(k)$ interpolating between two vacua of different *colours* $\mu \neq \nu$, as shown in Fig. 1. Similarly to the Ising model, the Potts model possesses a Kramers-Wannier duality $g \leftrightarrow 1/g$ connecting the two phases and their respective excitations [40–42].

When all the longitudinal fields $h_{\mu}$ are set to zero, the 3-state Potts model exhibits a global $\mathbb{S}_3$ permutation symmetry. The group has two independent generators $\mathcal{C}$ and $\mathcal{T}$, satisfying

$$\mathcal{T}^3 = \mathbb{I}, \qquad \mathcal{C}^2 = \mathbb{I}, \qquad \mathcal{C}\mathcal{T}\mathcal{C}^{-1} = \mathcal{T}^{-1} \,. \tag{3}$$

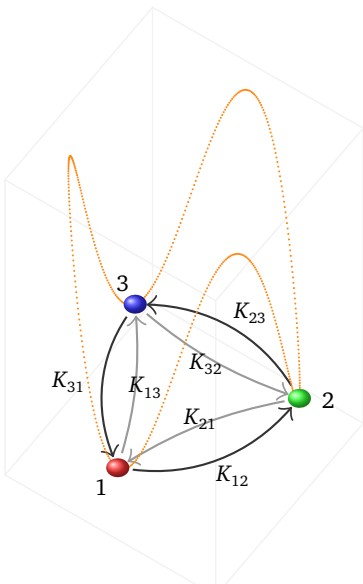

Figure 1: Vacuum configuration of the purely transverse 3-state Potts model in the broken phase ($g < 1$). The three colours (*red*, *green* and *blue*) are interpolated by kinks $K_{\mu\nu}$ with $\nu = \mu + 1$ mod 3 (counterclockwise) and anti-kinks $K_{\nu\mu}$ (clockwise) indicated by black and grey arrows respectively.

In the paramagnetic phase, one can introduce a basis in the magnonic space with one-particle states at fixed momentum given by $|A(k)\rangle$ and $|\bar{A}(k)\rangle$, transforming under the two-dimensional irreducible representation of $\mathbb{S}_3$ defined by the relations:

$$\mathcal{T}|A(k)\rangle = e^{\frac{2\pi i}{3}}|A(k)\rangle\,, \qquad \mathcal{T}|\bar{A}(k)\rangle = e^{-\frac{2\pi i}{3}}|\bar{A}(k)\rangle\,, \qquad \mathcal{C}|A(k)\rangle = |\bar{A}(k)\rangle\,, \tag{4}$$

and one can introduce the quasiparticle basis corresponding to the eigenstates of $\mathcal{C}$

$$|A_\pm(k)\rangle = \frac{1}{\sqrt{2}}\big(|A(k)\rangle \pm |\bar{A}(k)\rangle\big)\,, \tag{5}$$

which are degenerate when all longitudinal fields are zero, but for a non-zero $h_\mu$ the degeneracy is lifted.

In the ferromagnetic phase, the action of the cyclic generator $\mathcal{T}$ corresponds to the permutation of the vacua $\mu \to \mu + 1$ mod 3 with the appropriate permutation of the kink excitation:

$$\mathcal{T} = T_1 T_2 \cdots T_L\,, \tag{6}$$

where $T_i$ acts on site $i$ as

$$T_i = \begin{pmatrix} 0 & 1 & 0 \\ 0 & 0 & 1 \\ 1 & 0 & 0 \end{pmatrix}. \tag{7}$$

In contrast, $\mathcal{C}$ can be chosen as any of the three non-cyclic permutations, corresponding to swapping two of the vacua.

Switching on one or more longitudinal magnetic fields $h_\mu$ leads to explicitly breaking the symmetry group $\mathbb{S}_3$. Switching on only one of them, say $h_1$, partially breaks the symmetry, leaving an unbroken $\mathbb{Z}_2$ subgroup generated by the transformation swapping the spin directions 2 and 3, leaving 1 intact. One can choose the generator $\mathcal{C}$ to correspond to the unbroken subgroup given by

$$\mathcal{C} = C_1 C_2 \cdots C_L\,, \tag{8}$$

where $C_i$ acts at site $i$ as

$$C_i = \begin{pmatrix} 1 & 0 & 0 \\ 0 & 0 & 1 \\ 0 & 1 & 0 \end{pmatrix}. \tag{9}$$

The consequences of the explicit breaking of the symmetry on the dynamics in the paramagnetic phase were studied in [38]. In contrast, the present work considers the implications of introducing a longitudinal field in the ferromagnetic phase.

## 2.2  Quench protocols

The following quench protocols define the non-equilibrium time evolution we study. The initial state is the pure ferromagnetic state along direction 1:

$$|\psi_0\rangle = \bigotimes_{i=1}^{L} |1\rangle_i . \tag{10}$$

We distinguish four different types of quenches. The first two correspond to time evolution with finite transverse and longitudinal magnetic fields along the direction of the initial magnetisation:

$$|\psi(t)\rangle = e^{-iH(h_1,g)t} |\psi_0\rangle , \tag{11}$$

where $H(h_1, g)$ is the the Hamiltonian (2) with $0 < g < 1$, $h_1 \neq 0$, and $h_2 = h_3 = 0$. These cases are directly analogous to those studied in the Ising model [4, 22, 23]. Depending on the sign of the magnetic field $h_1$, there are two cases:

- **Positively aligned: $h_1 > 0$**
  The positive longitudinal field $h_1$ disfavours the magnetisation directions 2 and 3, corresponding to a pair of degenerate meta-stable (false) vacua with the true vacuum corresponding to direction 1 as shown in Fig. 2a.

- **Negatively aligned: $h_1 < 0$**
  Now magnetisation directions 2 and 3 are favoured over 1, leading to a pair of degenerate true vacua, while direction 1 corresponds to a meta-stable (false) vacuum state as shown in Fig. 2b.

Note that the kinks and antikinks interpolating between the false vacua if $h_1 > 0$ ($K$ and $\bar{K}$ in Fig. 2a), alternatively between the true vacua if $h_1 < 0$ ($K$ and $\bar{K}$ in Fig. 2b) realise an effective Ising subsystem, and can be well approximated by free fermions.

Unlike the Ising model, the Potts model allows quenching with a longitudinal field that is neither parallel nor anti-parallel to the initial state. We call this case an 'oblique' quench which determines the other two protocols by the following time evolution:

$$|\psi(t)\rangle = e^{-iH(h_2,g)t} |\psi_0\rangle , \tag{12}$$

where $H(h_2, g)$ is the the Hamiltonian (2) with $0 < g < 1$, $h_2 \neq 0$, and $h_1 = h_3 = 0$.

- **Positive oblique: $h_2 > 0$**
  The initial state is in the direction of the first of the two degenerate false vacua 1 and 3, while direction 2 corresponds to the true vacuum of the theory as shown in Fig. 3a.

- **Negative oblique: $h_2 < 0$**
  The initial state is in the direction of the first of the two degenerate true vacua 1 and 3, while direction 2 corresponds to the false vacuum as shown in Fig. 3b.

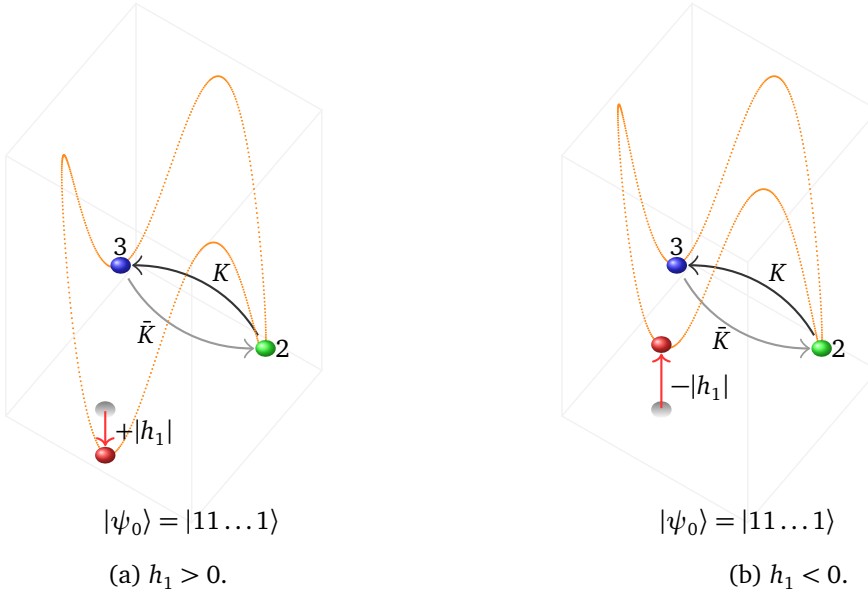

$$|\psi_0\rangle = |11\ldots1\rangle \qquad\qquad |\psi_0\rangle = |11\ldots1\rangle$$

(a) $h_1 > 0$.        (b) $h_1 < 0$.

Figure 2: Vacuum configuration with a longitudinal field $h_1$. Vacua 2 and 3 with colours *green* and *blue* remain degenerate. Depending on the sign of $h_1$, they are either metastable or stable, with kink $K$ and antikink $\bar{K}$ excitations interpolating between the two. The initial state $|\psi_0\rangle$ is polarised along the *red* direction.

These protocols explore different ways to break explicitly the $\mathbb{S}_3$ permutation symmetry while leaving a subgroup $\mathbb{Z}_2$ unbroken. We emphasise once again that the spectra of the Hamiltonian governing the positively aligned and the positive oblique quenches and the Hamiltonian determining the time evolution of the negatively aligned and the negative oblique quenches are the same, respectively. However, the aligned and oblique cases differ markedly due to the orientation of the polarisation of the initial state relative to the longitudinal field. In the aligned cases, it is parallel/anti-parallel compared to the longitudinal field, while in the oblique cases, it points in a different direction.

## 2.3 Quench phenomenology

We study the time evolution of the spin chain in the thermodynamic limit $L = \infty$, which is computed using the infinite Time Evolving Block Decimation (iTEBD) algorithm [43] with second-order Trotterisation. In this section, we include numerical results that were produced with maximal bond dimension $\chi_{\max} = 300$ and time step $\delta = 0.005$. We give further details on our numerical simulations in Appendix C. Due to translational invariance, expectation values of local operators are position-independent, while correlation functions only depend on the relative position.

The phenomenology of the quenches can be studied via the following observables:

- Magnetisation: there are three different magnetisations defined as

$$M_\mu(t) = \langle\psi(t)|P^\mu|\psi(t)\rangle. \tag{13}$$

However, they are not independent as they sum to zero:

$$\sum_{\mu=1}^{3} M_\mu(t) = 0. \tag{14}$$

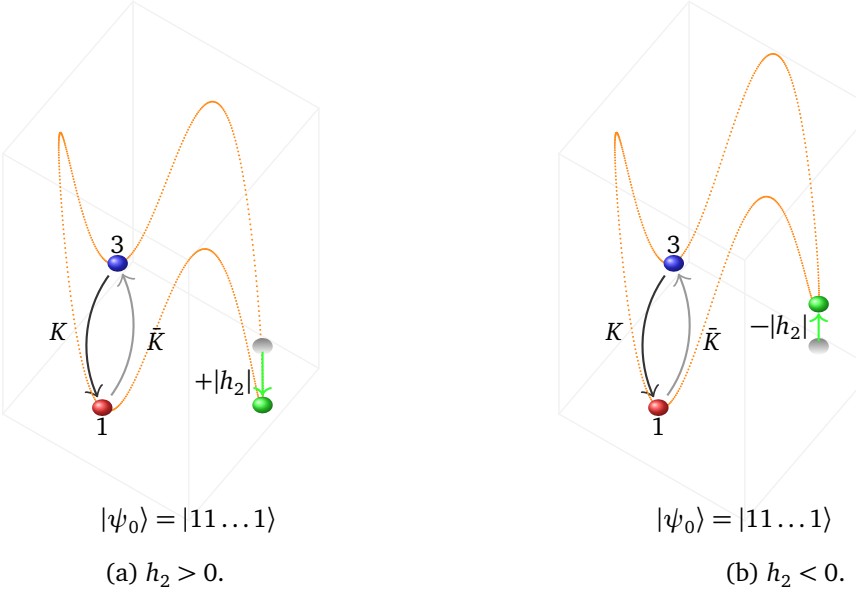

|ψ₀⟩ = |11...1⟩           |ψ₀⟩ = |11...1⟩

(a) $h_2 > 0$.                  (b) $h_2 < 0$.

Figure 3: Vacuum configuration with a longitudinal field $h_2$. Vacua 1 and 3 corresponding to colours *red* and *blue* remain degenerate. Depending on the sign of $h_2$, they are either metastable or stable, with kink $K$ and antikink $\bar{K}$ excitations interpolating between the two. As in Fig. 2, the initial state $|\psi_0\rangle$ is polarised along the *red* direction.

We choose $M_1$ and $M_2$ as independent magnetisations, while $M_3$ can be computed from (14) as

$$M_3 = -M_1 - M_2 \,. \tag{15}$$

If the $\mathbb{Z}_2$ subgroup generated by $\mathcal{C}$ of (8) is preserved, we have the further constraint $M_2 = M_3$ and they can be computed from the only independent magnetisation $M_1$ as

$$M_2 = M_3 = -\frac{M_1}{2} \,. \tag{16}$$

- Correlation functions (connected):

$$C_{\mu\nu}(l,t) = \langle\psi(t)| P_l^\mu P_0^\nu |\psi(t)\rangle - M_\mu(t) M_\nu(t) \,. \tag{17}$$

Since $[P_i^\mu, P_i^\nu] = 0$ they are symmetric ($C_{\mu\nu} = C_{\nu\mu}$) and satisfy further constraints similar to (14):

$$\sum_{\mu=1}^3 C_{\mu\nu} = 0 \,. \tag{18}$$

Without further symmetries, there are then three independent correlators. We choose $C_{11}, C_{22}$ and $C_{23}$ as such, while the others can be computed from (18) as

$$C_{12} = -C_{22} - C_{23} \,, \qquad C_{13} = -C_{11} + C_{22} + C_{23} \,, \qquad C_{33} = C_{11} - C_{22} - 2C_{23} \,. \tag{19}$$

With the further constraint $C_{22} = C_{33}$ imposed by the $\mathbb{Z}_2$ symmetry, we choose $C_{11}$ and $C_{23}$ as independent and

$$C_{12} = -\frac{C_{11}}{2} \,, \qquad C_{13} = C_{12} = -\frac{C_{11}}{2} \,, \qquad C_{22} = C_{33} = \frac{C_{11}}{2} - C_{23} \,. \tag{20}$$

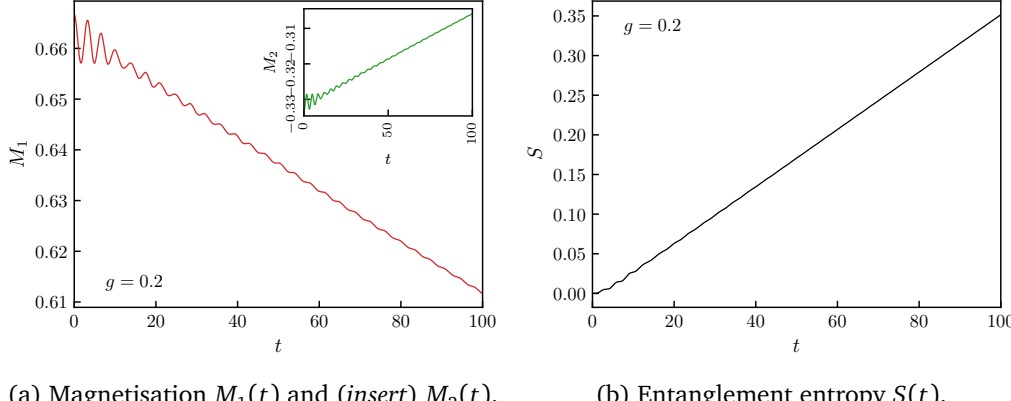

(a) Magnetisation $M_1(t)$ and (*insert*) $M_2(t)$.      (b) Entanglement entropy $S(t)$.

Figure 4: Time evolution of magnetisations $M_1$ and $M_2$ (*left*) and the linearly growing entanglement entropy (*right*) in the pure transverse field quench with transverse field $g = 0.2$.

- Entanglement entropy: the von Neumann entropy of a subsystem $A$ is defined as

$$S_A(t) = -\operatorname{Tr} \rho_A(t) \ln \rho_A(t),\tag{21}$$

where the reduced density matrix is the partial trace over the complement $\bar{A}$ of the subsystem $A$:

$$\rho_A(t) = \operatorname{Tr}_{\bar{A}} |\psi(t)\rangle \langle \psi(t)|.\tag{22}$$

Here, we consider the case when $A$ is one (semi-infinite) half of the system.

We now consider quenches in the Potts model, choosing the transverse field of the post-quench Hamiltonian deep in the ferromagnetic phase $g < 1$ and a suitably small longitudinal field. We start with the pure transverse quench as a warm-up and then turn to the four quench protocols introduced in Subsection 2.2.

### 2.3.1 Pure transverse quench in the ferromagnetic phase

For quenches with pure transverse field, the initial magnetisation is expected to decay to zero exponentially, following the same behaviour observed in the Ising spin chain [44, 45]. The reason is that the quench results in a state of finite energy density, which is later expected to equilibrate to a finite temperature. However, a discrete symmetry cannot be spontaneously broken at nonzero temperatures in a one-dimensional system with short-range interactions. The initial state (10) leaves unbroken the permutation symmetry $\mathbb{Z}_2$ between magnetisation directions 2 (*green*) and 3 (*blue*), therefore there is a single independent magnetisation $M_1$, and only two independent correlation functions.

The evolution of magnetisation $M_1$ is shown in Fig. 4a, which covers the initial part of the relaxation and is indeed very similar to the behaviour observed in the Ising spin chain [44, 45]; the evolution of $M_2$ shown in the inset is not independent. The linear growth of the entanglement entropy is also fully analogous to the behaviour in the quantum Ising spin chain [46], which is explained by the semiclassical quasiparticle picture of post-quench time evolution.

The correlation functions show simple light-cone behaviour, which is again explained by the semiclassical quasiparticle picture [28, 29]. The slope of the displayed lines is determined by the Lieb-Robinson velocity of the kinks $v_{LR}$ given by (A.5), which can be computed from

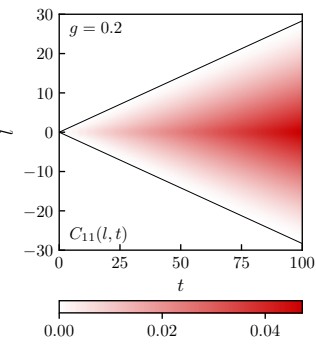
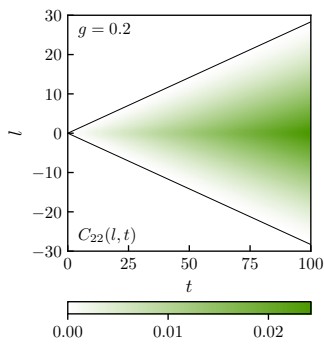
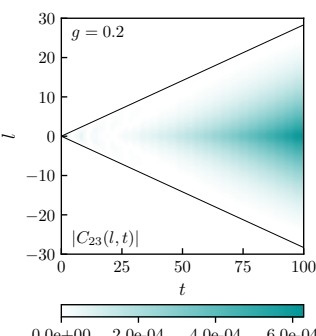

(a) Connected 11 (*red-red*) correlations.

(b) Connected 22 (*green-green*) correlations.

(c) Connected 23 (*cyan*) correlations.

Figure 5: The light-cone behaviour of correlation functions in the pure transverse field quench with $g = 0.2$. Solid black lines are drawn at $\pm 2v_{LR}t$, with $v_{LR}$ being the maximum velocity of kinks given by (A.5).

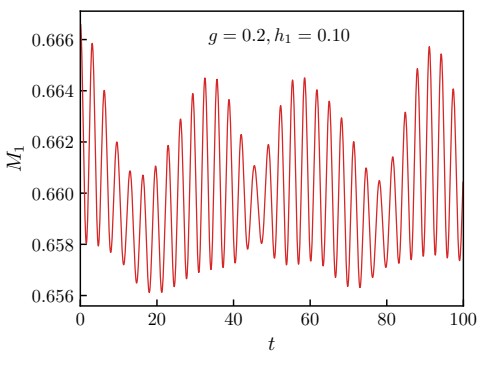
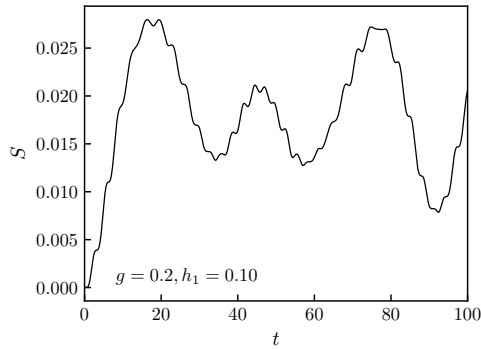

(a) Magnetisation $M_1(t)$.

(b) Entanglement entropy $S(t)$.

Figure 6: The time evolution of magnetisation $M_1$ (*left*) shows persistent oscillations and the entanglement entropy (*right*) saturates after a short initial period of time in the positively aligned regime with transverse field $g = 0.2$ and longitudinal field $h_1 = 0.10$.

the dispersion relation determined using exact diagonalisation as discussed in Appendix A. Although only two of them are independent, we display $C_{11}$, $C_{22}$ and $C_{23}$ to facilitate comparison with quenches in the presence of longitudinal fields.

### 2.3.2 Positively aligned

In this protocol, we start with the state (10), which is polarised along the *red* direction 1 and introduce a post-quench longitudinal field $h_1 > 0$. The remaining $\mathbb{Z}_2$ symmetry between directions 2 and 3 imposes $M_2(t) = M_3(t)$, therefore (14) implies

$$M_1(t) = -2M_2(t) = -2M_3(t), \tag{23}$$

so only one magnetisation is independent. Fig. 6 shows the evolution of magnetisation, as well as the entanglement entropy, which is generated by the Hamiltonian (2) with longitudinal fields $h_1 \neq 0$ and $h_2 = h_3 = 0$.

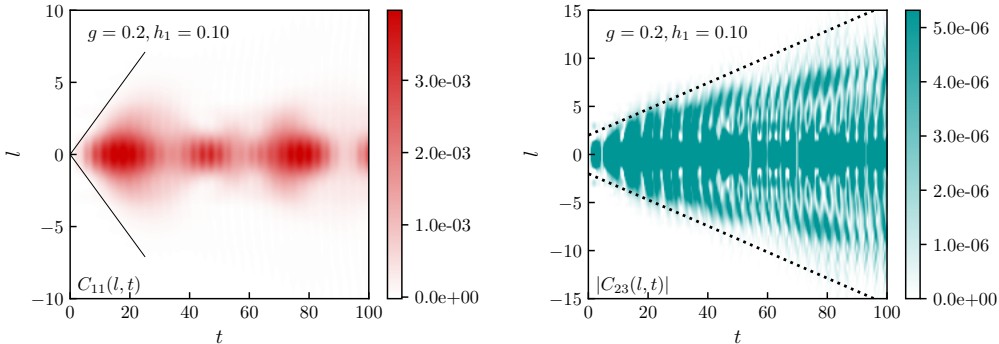

(a) Connected 11 (*red-red*) correlations.     (b) Connected 23 (*cyan*) correlations.

Figure 7: Time evolution of correlation functions in the positively aligned regime with transverse field $g = 0.2$ and longitudinal field $h_1 = 0.10$. $C_{11}(l, t)$ *red-red* correlations are confined (*left*). Here, solid black lines mark the light cone in the pure transverse quench shown in Fig. 5. In contrast, the *green-blue (cyan)* correlator $C_{23}(l, t)$ shows escaping fronts, with dotted black lines showing the corresponding light cone (*right*).

Clearly, switching on a longitudinal field radically alters the dynamics compared to the pure transverse case. As demonstrated in Fig. 6a, the magnetisation shows persistent oscillations instead of exponential relaxation, while Fig. 6b shows that the growth of entanglement entropy saturates after a short initial period and it also starts oscillating. This is fully analogous to the behaviour observed in the Ising case [4] and has the same explanation by dynamical confinement. As we discuss in detail later, the frequencies of the observed oscillations correspond to the 'hadron' spectrum resulting from confinement. The dominant effect comes from *mesons,* which are bound states of two kinks formed under the effect of the attractive linear potential due to the presence of the longitudinal field [25], in full analogy with the Ising case [47]. However, the three states of magnetisation of the Potts model, which are analogous to the three different 'colours' of quantum chromodynamics, allow for the formation of *baryons* as well [27]. They appear in the oscillation spectrum as demonstrated in Subsection 4.1.1.

Another characteristic signature of dynamical confinement is the suppression of the spreading of correlation. For the quench considered here, only two of the correlation functions are independent, as stated at the beginning of the section. Their time evolution is displayed in Fig. 7. The $C_{11}(l, t)$ correlator shows the characteristic suppression of the light-cone propagation of correlations resulting from dynamical confinement [4]. This is demonstrated clearly by the deviation from the behaviour of the pure transverse quench, which is due to the localisation of kink excitation by the confining force.

However, $C_{23}(l, t)$ shows the presence of residual light-cone structure indicated by the dotted black lines in Fig. 7b. Due to the polarisation of the initial state, the degrees of freedom localised in the directions 2 and 3 are strongly suppressed, which is reflected in the 23 correlations being two orders of magnitude smaller than 11 correlations. Nevertheless, the residual spreading of correlations is also reflected in the entanglement entropy, which shows a drift over long time scales (cf. Fig. 39a). We return to a further discussion of this phenomenon in Subsection 4.2.1.

### 2.3.3 Negatively aligned

When reversing the sign of $h_1$ compared to the previous case, the initial state now corresponds to the false vacuum, while the system has two true vacua as shown in Fig. 2a. This

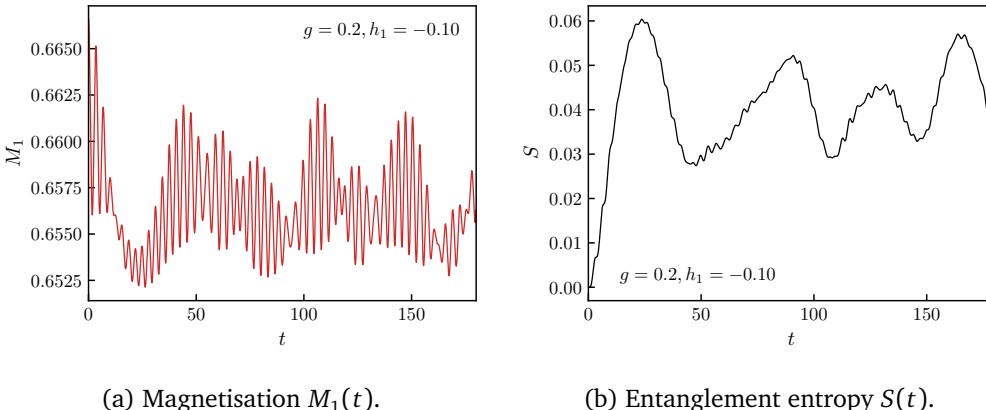

(a) Magnetisation $M_1(t)$.      (b) Entanglement entropy $S(t)$.

Figure 8: The time evolution of magnetisation $M_1$ (*left*) shows persistent oscillations and the entanglement entropy (*right*) saturates after a short initial period of time in the negatively aligned regime with transverse field $g = 0.2$ and longitudinal field $h_1 = -0.10$.

is the classical set-up of the decay of the false vacuum [20, 21], apart from the degeneracy of the true vacuum (similar to some scenarios considered recently in [48]). However, contrary to the case of continuum quantum field theory, the vacuum decay is suppressed by Bloch oscillations [22], i.e., the expansion of nucleated bubbles of the true vacuum is prevented by Stark localisation [10]. This results in features similar to dynamical confinement, such as persistent oscillation of the magnetisation $M_1(t)$ shown in Fig. 8 (*left*), early termination of the growth of entanglement entropy (Fig. 8 - *right*) and suppression of the light-cone spreading of correlations (Fig. 9).

However, despite the qualitative similarity of the time evolution to the positively aligned (confining) case, the different underlying mechanism leads to quantitative differences. As shown later in Subsection 4.1.2, the spectrum of oscillations is now dominated by different excitations corresponding to bubbles of true vacuum nucleated in the initial false vacuum state, as observed previously in the Ising case [22, 23, 49].

As in the previous case, $C_{23}(l, t)$ shows that even though directions 2 and 3 now correspond to true vacua, correlations in this channel are two orders of magnitude smaller due to the initial polarisation pointing in direction 1. Nevertheless, just like in the positively aligned (confining) quench, there is still a residual light cone indicated by dotted lines, discussed further in Subsection 4.2.1. In addition, the entanglement entropy again shows a drift over long time scales, as demonstrated in Fig. 39b.

### 2.3.4 Positive oblique

Now we consider time evolution with a longitudinal magnetisation pointing in a direction different from the polarisation of the initial state (10), as specified in (12). First, we discuss this oblique quench scenario for the case when $h_2$ is positive. The Hamiltonian (2) with longitudinal fields $h_2 \neq 0$ and $h_1 = h_3 = 0$ is invariant under the unbroken subgroup $\mathbb{Z}_2$ between colours 1 and 3, but the initial state breaks this symmetry, resulting in two independent magnetisations $M_2$ and $M_3$. Their time evolution is shown in Fig. 10 along with the entanglement entropy one. While $M_2$ shows persistent oscillations, $M_3$ shows an increasing trend similar to the pure transverse quench (c.f. Fig. 4a). This behaviour, as well as the unsuppressed growth of entanglement entropy, strongly suggests that the corresponding degrees of freedom relax.

This spreading of correlations is also observed in the correlation function $C_{11}(l, t)$, where the standard light cone structure is restored. In contrast, light-cone spreading in $C_{22}(l, t)$

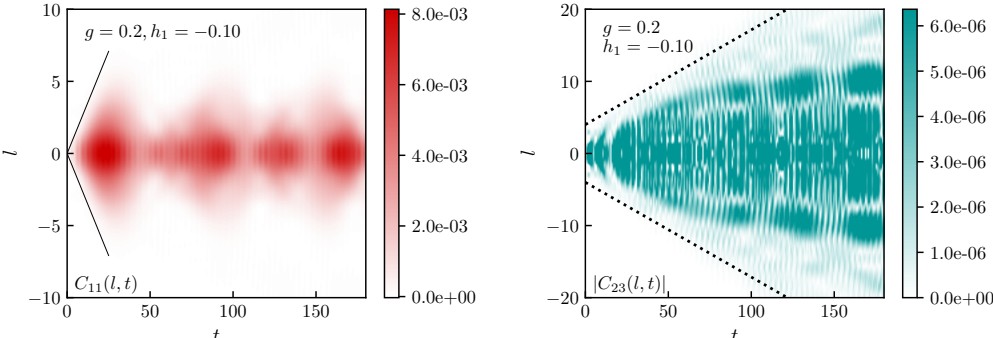

(a) Connected 11 (*red-red*) correlations.      (b) Connected 23 (*cyan*) correlations.

Figure 9: Time evolution of correlation functions in the negatively aligned regime with transverse field $g = 0.2$ and longitudinal field $h_1 = -0.10$. $C_{11}(l, t)$ (*left*) shows localised *red-red* correlations, with solid black lines marking the light cone in the pure transverse quench shown in Fig. 5. Escaping fronts in the *green-blue* (*cyan*) correlator $C_{23}(l, t)$ are present, with dotted black lines showing the corresponding light cone (*right*).

remains suppressed (Fig. 11). As a result, the phenomenology of this quench is mixed: while in direction 2 (*green*), a behaviour similar to confinement or anticonfinement is observed due to the initial polarisation in direction 1 (*red*), other degrees of freedom excited in the quench show relaxation and light-cone spreading of correlations, leading to unsuppressed growth of entanglement entropy. Finally, correlation $C_{23}(l, t)$ is suppressed altogether, although it does display a clear light cone structure. We discuss this behaviour further in Subsection 4.2.2.

### 2.3.5 Negative oblique

In this case, $h_2$ is negative, and we observe qualitatively the same features as for $h_2 > 0$. However, the relaxation in $M_3$ as well as the growth of entanglement entropy both appear faster than in the $h_2 > 0$ case, as demonstrated in Fig. 12.

Fig. 13 shows the time evolution of correlation functions. We note that the light-cone spreading, as well as the magnitude of the correlation $C_{22}(l, t)$, is more suppressed spatially compared to the case $h_2 > 0$. In contrast, the spreading of correlations shown by $C_{11}(l, t)$ appears to be slightly enhanced. We discuss these features further in Subsection 4.2.2.

## 3 Semiclassical description of the spectrum

Here we construct the semiclassical particle content of the theory. Switching on a longitudinal field $h_\mu$ induces a linear potential between kinks, which enclose a domain of length $d$ with magnetisation different from $\mu$, with the magnetisation outside the enclosed domain polarised in the $\mu$ direction. For small values of the field, the potential can be well approximated as

$$V(d) = \pm \chi d, \tag{24}$$

where $\chi = |h_\mu|(M_\mu - M_{\nu \neq \mu})$ is the string tension, the sign is chosen to reflect the sign of $h_\mu$, and the spontaneous magnetisations $M_\alpha$ are computed for the pure transverse case with all $h_\alpha = 0$ and a ground state polarised in the $\mu$ direction (cf. Appendix C). Using the results of Appendix A.4 we have

$$\chi = |h_\mu|(1 - g^2)^\alpha, \tag{25}$$

with $\alpha \approx 0.102$.

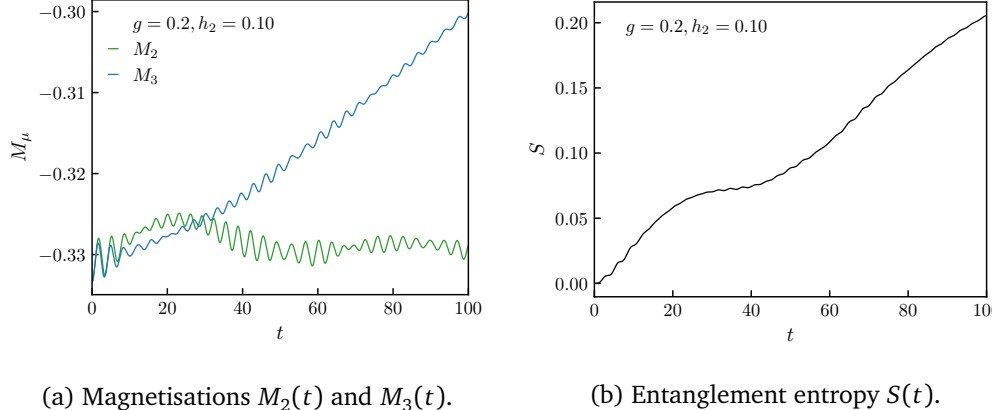

(a) Magnetisations $M_2(t)$ and $M_3(t)$.    (b) Entanglement entropy $S(t)$.

Figure 10: Magnetisations $M_2(t)$, exhibiting persistent oscillations, and $M_3(t)$, showing an increasing trend (*left*), and the time evolution of the entanglement entropy (*right*) displaying unsuppressed growth in the positive oblique regime with transverse field $g = 0.2$ and longitudinal field $h_2 = 0.10$.

In contrast with the Ising chain, where the kinks are non-interacting when the longitudinal field vanishes, the Potts kinks have a short-range interaction resulting in non-trivial scattering [40]. We first treat the problem by neglecting this interaction and derive a semiclassical description of the spectrum of two-kink states by only considering the linear potential (24). We label this case 'non-interacting kinks' despite the presence of the linear potential; strictly speaking, it is only non-interacting when the longitudinal field is also switched off. The cases of positive ($h_\mu > 0$) and negative ($h_\mu < 0$) longitudinal fields are considered in Subsections 3.1 and 3.2. The effect of the short-range interaction is taken into account at a later stage in Subsection 3.3. The semiclassical results are validated by comparing them to exact diagonalisation results, which we summarise in Appendix B.

We note that the semiclassical approach considered below assumes that the kinks can be treated as point-like, which requires the transverse magnetic field $g$ not to be too close to the quantum phase transition point $g_c = 1$. Additionally, the typical distance between the kinks must be large, corresponding to weak confinement with $\chi$ less than order one [4, 50].

### 3.1 Non-interacting kinks over a true vacuum: Mesons

For $h_\mu > 0$, the linear potential (24) has a positive sign; therefore, it is attractive and induces kink confinement.

#### 3.1.1 Classical trajectories

Consider two kinks moving classically in one dimension with the potential (24):

$$\mathcal{H} = \epsilon(k_1) + \epsilon(k_2) + \chi|x_2 - x_1|, \tag{26}$$

where $(k_1, k_2)$ are the momenta of the single kinks, $(x_1, x_2)$ are their positions and $\epsilon(k)$ is given by the kink dispersion relation for the pure transverse Potts chain ($\{h_\mu = 0\}$) that can be computed by exact diagonalisation as discussed in Appendix A. After the canonical transformation

$$X = \frac{x_1 + x_2}{2}, \qquad x = x_2 - x_1, \qquad K = k_1 + k_2, \qquad k = \frac{k_2 - k_1}{2}, \tag{27}$$

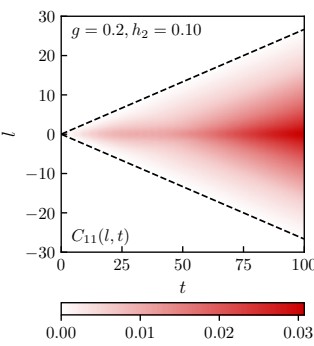

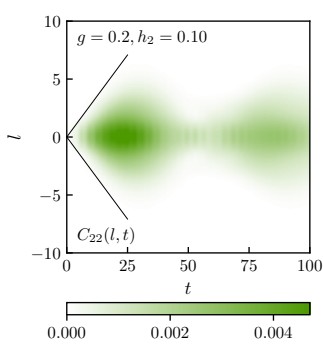

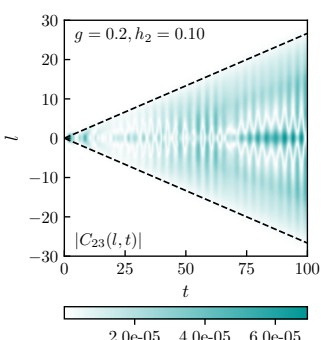

(a) Connected 11 (*red-red*) correlations.

(b) Connected 22 (*green-green*) correlations.

(c) Connected 23 (*cyan*) correlations.

Figure 11: Time evolution of correlation functions in the positive oblique regime with transverse field $g = 0.2$ and longitudinal field $h_2 = 0.10$. The *red-red* correlators $C_{11}(l, t)$ (*left*) and the *cyan* correlator $C_{23}(l, t)$ (*right*) show well-defined light cones delimited by dashed black lines, although the magnitude of $C_{23}$ is strongly suppressed. In contrast, the *green-green* correlator $C_{22}(l, t)$ is confined (*middle*). Here, the solid black lines mark the light cone in the corresponding pure transverse quench shown in Fig. 5.

the Hamiltonian becomes

$$\mathcal{H} = \omega(k; K) + \chi|x|, \quad \text{where} \quad \omega(k; K) = \epsilon(K/2 + k) + \epsilon(K/2 - k), \tag{28}$$

resulting in the canonical equations of motion

$$\dot{X}(t) = \frac{\partial \omega(k; K)}{\partial K}, \qquad \dot{K}(t) = 0, \qquad \dot{x}(t) = \frac{\partial \omega(k; K)}{\partial k}, \qquad \dot{k}(t) = -\chi \operatorname{sign}(x(t)). \tag{29}$$

For a given value of the total momentum $K$, these equations describe the relative motion of two particles, which strongly depends on the values of the conserved total momentum $K$ and total energy $E = \omega(k; K) + \chi|x|$. Our analysis below follows closely the one presented in [51] for the XXZ spin chain.

○ Regime (I)

For $K < K_c$, with

$$K_c = 2 \arccos\left(\frac{-A + \sqrt{A^2 - B^2}}{B}\right), \tag{30}$$

$\omega(k; K)$ is a single-well potential and equation $\omega(k; K) = E$ has two solutions $k = \pm k_a$ when $\omega(0; K) < E < \omega(\pi; K)$, bounding the kinematically allowed region (see Fig. 14, *left*). Choosing the initial conditions as $x(0) = 0$ and $k(0) = k_a > 0$, the momentum $k$ linearly decreases in time $k = k_a - \chi t$ until $t_1 = 2k_a/\chi$. In turn, the spatial coordinate $x$ increases up to its maximal value

$$x_{\max} = \frac{E - \omega(0; K)}{\chi}, \tag{31}$$

taken at $t = t_1/2$ and then decreases until reaching its initial zero value at $t = t_1$ when the kinks collide. After the collision we have $k(t) = -k_a + \chi t$, until $k$ returns to its initial

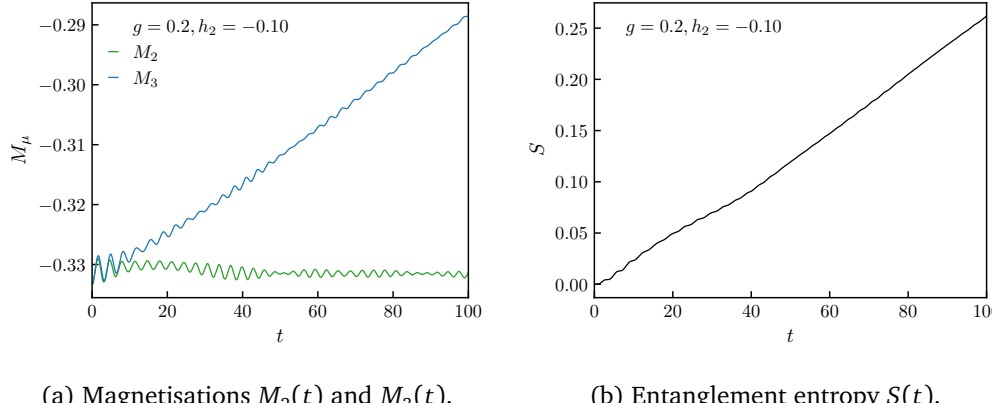

(a) Magnetisations $M_2(t)$ and $M_3(t)$.   (b) Entanglement entropy $S(t)$.

Figure 12: Magnetisations $M_2(t)$, exhibiting persistent oscillations, and $M_3(t)$, showing an increasing trend (*left*), and the time evolution of the entanglement entropy displaying unsuppressed growth (*right*) in the negative oblique regime with transverse field $g = 0.2$ and longitudinal field $h_2 = -0.10$.

value $k(2t_1) = k_a$. The resulting motion shown in Fig. 14 (*right*) has a period $2t_1$. The bound state drifts with an average velocity of

$$\left\langle \dot{X} \right\rangle = \frac{1}{2t_1} \int_0^{2t_1} dt\, \dot{X}(t) = \frac{\epsilon(k_{2a}) - \epsilon(k_{1a})}{k_{2a} - k_{1a}}, \quad \text{where} \quad k_{1a} = \frac{K}{2} - k_a, k_{2a} = \frac{K}{2} + k_a. \quad (32)$$

○ Regime (II)

At higher energies $E > \omega(\pi; K)$, the kinematically allowed regions for momentum $k$ is the whole real axis and, assuming $x(0)$ positive, $k$ linearly decreases in time $k(t) = k(0) - \chi t$, while the coordinate $x$ oscillates between $x_{\max}$ given by (31) and

$$x_{\min} = \frac{E - \omega(\pi; K)}{\chi}. \quad (33)$$

Since both $x_{\min}$ and $x_{\max}$ are positive, the two kinks never meet and undergo Bloch oscillations resulting in collisionless mesons. The period of oscillation is $t_2 = 2\pi/\chi$, and the kinks do not drift along the chain since $\left\langle \dot{X} \right\rangle = 0$. Their spatial coordinates can be evaluated as

$$x_1(t) = x_{10} + \frac{1}{\chi}[\epsilon(k_{10} + \chi t) - \epsilon(k_{10})], \qquad x_2(t) = x_{20} - \frac{1}{\chi}[\epsilon(k_{20} - \chi t) - \epsilon(k_{20})], \quad (34)$$

with $k_{i0} = k_i(0)$, $x_{i0} = x_i(0)$. Fig. 15 shows the Bloch oscillations of the two particles in real space.

○ Regime (III)

The last dynamical regime is when the potential $\omega(k; K)$ develops a double well with a local maximum at $k = 0$ as shown in Fig. 16 (*left*). This happens for $K > K_c$, and

$$\min_k \omega(k; K) < E < \omega(0; K). \quad (35)$$

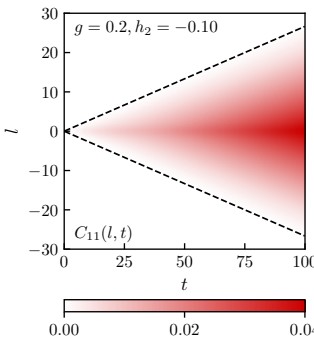

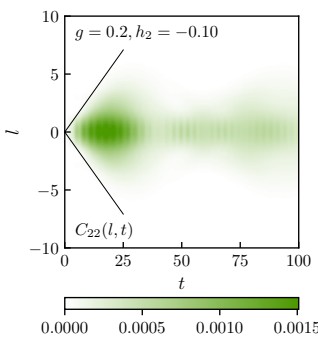

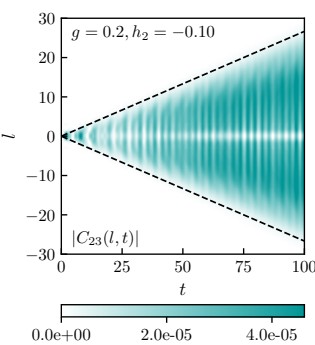

(a) Connected 11 (*red-red*) correlations.

(b) Connected 22 (*green-green*) correlations.

(c) Connected 23 (*cyan*) correlations.

Figure 13: Time evolution of correlation functions in the negative oblique regime with transverse field $g = 0.2$ and longitudinal field $h_2 = -0.10$. The *red-red* correlator $C_{11}(l, t)$ (*left*) and the *cyan* correlator $C_{23}(l, t)$ (*right*) show unsuppressed light cones delimited by dashed black lines, although the magnitude of $C_{23}$ is strongly suppressed. In contrast, the *green-green* correlator $C_{22}(l, t)$ is confined (*middle*). Here the solid black lines mark the light cone in the corresponding pure transverse quench shown in Fig. 5.

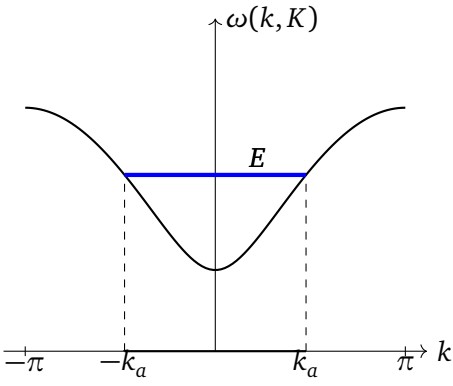

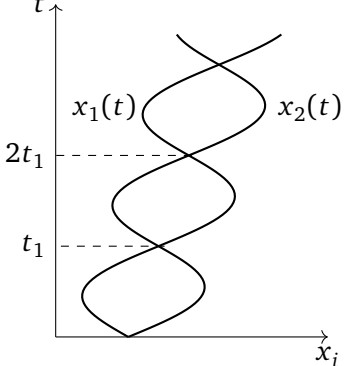

Figure 14: Potential and kink trajectories in the first dynamical regime.

In this case, the kinematically allowed region consists of two disconnected pieces, since the equation $E = \omega(k; K)$ has four solutions $\{\pm k_b, \pm k_a\}$ with $0 < k_b < k_a$. Starting from $k(0) = k_a$ and $x(0) = 0$, the motion is restricted to one of the wells, and the solution is given by

$$k(t) = k_a - \chi t, \quad \text{for} \quad 0 < t < t_3 = \frac{k_a - k_b}{\chi}, \tag{36}$$

where $k(t_3) = k_b$ and $x(t_3) = 0$. After the collision $k(t) = k_b + \chi t$, until $k$ returns to its initial value $k(2t_3) = k_a$, therefore the momentum $k(t)$ and the spatial coordinate $x(t)$ are periodic functions of period $2t_3$. Finally, the bound state drifts with the average velocity

$$\langle \dot{X} \rangle = \frac{\epsilon(k_{2a}) - \epsilon(k_{1a}) - \epsilon(k_{1a}) + \epsilon(k_{1b})}{2(k_{1b} - k_{1a})}, \quad \text{where} \quad k_{1b} = \frac{K}{2} - k_b, \, k_{2a} = \frac{K}{2} + k_b. \tag{37}$$

The kink trajectories in this dynamical regime are shown in Fig. 16 (*right*).

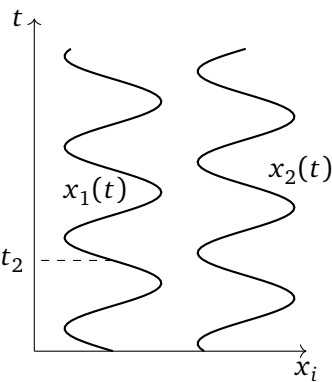

Figure 15: Kink trajectories in the second dynamical regime giving rise to Bloch oscillations.

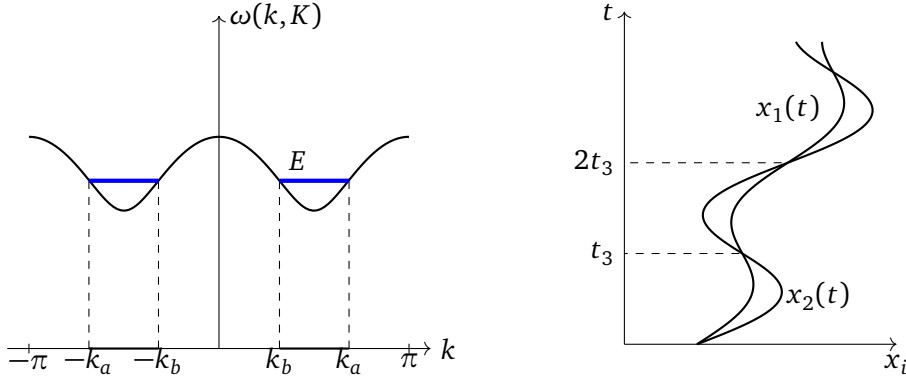

Figure 16: Potential and kink trajectories in the third dynamical regime.

### 3.1.2 Semiclassical quantisation of mesons

The meson configurations described above can be quantised using Bohr-Sommerfeld quantisation. For the first dynamical regime (I) the appropriate condition takes the form

$$\oint dx\, k(x) = 2\pi(\nu + 1/2), \tag{38}$$

where the $1/2$ results from the presence of the turning points, and the quantum number $\nu = 0, 1, 2, \dots$ is the number of wave function nodes. As shown in [47], Eq. (38) leads to the quantisation condition

$$4 E_\nu(K)\, k_a - 2 \int_{-k_a}^{k_a} dk\, \omega(k; K) = 2\pi\chi\left(\nu + \frac{1}{2}\right), \tag{39}$$

where the turning point $k_a$ is determined by

$$E_n(K) = \omega(k_a; K), \tag{40}$$

and $E_\nu(K)$ is the energy of the meson state. The quantum number $\nu$ can be parameterised for odd wave functions as $\nu = 2n - 1$, while for even wave functions $\nu = 2n - 2$ with $n = 1, 2, \dots$

Therefore the semiclassical quantisation can be rewritten as

$$2E_n^{(\kappa)}(K)k_a - \int_{-k_a}^{k_a} dk\,\omega(k;K) = 2\pi\chi\left(n - \frac{1}{2} + \frac{(-1)^\kappa}{4}\right), \tag{41}$$

where $\kappa = 0$ (1) corresponds to spatially odd (even) states. We remark that in the Ising case considered in [47] only the odd wave functions are physical due to the fermionic nature of kink excitations. However, in the Potts case, both even and odd wave functions correspond to physical states due to the presence of colours since the internal structure of a neutral two-kink state allows both singlet and triplet as discussed in Appendix A.3.1; c.f. also Subsection 3.3. The same happens in the scaling limit considered in [25].

In the dynamical regime (II) there are no turning points for the momentum $k(t)$, and the periodicity of the wave function in the Brillouin zone $-\pi \le k < \pi$ results in the condition

$$2\pi E_n(K) - \int_{-\pi}^{\pi} dk\,\omega(k;K) = 2\pi\chi n. \tag{42}$$

Using the periodicity of $\epsilon(k)$ gives energy levels independent of the total momentum $K$:

$$E_n = n\chi + \frac{1}{\pi}\int_{-\pi}^{\pi} dk\,\epsilon(k), \tag{43}$$

with their spacing given by the angular frequency $\chi$ of the Bloch oscillations.

In the dynamical regime (III), the semiclassical motion occurs in one of two separated wells with turning points $\pm k_a$ and $\pm k_b$ with $k_a > k_b$. Since this interval is not invariant under the reflection $k \to -k$, we do not distinguish between odd and even cases hence we obtain the single expression

$$E_n(K)(k_a - k_b) - \int_{k_b}^{k_a} dk\,\omega(k;K) = \pi\chi\,(n - 1/2)\,. \tag{44}$$

In our subsequent calculations, we do not encounter this case, so we omit it in the sequel.

## 3.2 Non-interacting kinks over a false vacuum: Bubbles

Choosing $h_\mu < 0$, the $\mu$-polarised ground state becomes a false vacuum. The low energy spectrum built upon this state corresponds to bubbles of the true vacuum [23, 49, 52]. The nucleation of these bubbles is an essential step in the standard scenario of vacuum decay [20]. However, contrary to the field theory case, in the lattice model, the subsequent expansion of the bubbles is prevented by Bloch oscillations [22] resulting in the localisation observed in the negatively aligned quenches discussed in Subsection 2.3.3. We note that the persistent oscillations induced by these bubbles provide useful signatures to detect and characterise metastable vacuum states [53, 54].

### 3.2.1 Classical trajectories

The effective two-kink Hamiltonian differs from (26) by the sign of the string tension:

$$\mathcal{H} = \omega(k;K) - \chi|x|, \tag{45}$$

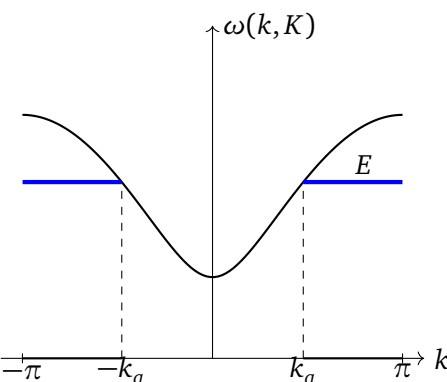

Figure 17: Potential and orbit (in blue) for bubbles in the first dynamical regime.

corresponding to a repulsive force between the kinks. Following [22], we call this the anti-confining case or simply refer to it as anticonfinement.

The canonical equations of motion are the same as in (29), but flipping the sign $\chi \to -\chi$ in the equation for $k$. Apart from that, the argument proceeds as before. For example in the Regime (I), by choosing

$$x(0) = 0, \qquad k(0) = k_a > 0, \tag{46}$$

due to (29), the momentum $k$ linearly increases in time $k = k_a + \chi t$ until

$$t = \frac{\tilde{t}_1}{2} = \frac{\pi - k_a}{\chi}, \tag{47}$$

where $k = \pi$. The classical orbit, shown in Fig. 17, is now *below* the $k$-space 'potential' function $\omega(k, K)$ and connects the turning point $k = k_a$ to the edge of the Brillouin zone $k = \pi$, crossing to $k = -\pi$ and then increasing again until $k = -k_a$. Then the kinks collide and the motion of the momentum is reversed until $t = 2\tilde{t}_1$ when it returns to its original value. Then, the cycle repeats again, describing the relative motion of two particles with a period $2\tilde{t}_1$.

In the dynamical regime (II), one can proceed in the same way as for the mesons: the kinematically allowed regions for momentum $k$ are the whole real axis. The two kinks never meet; instead, they undergo Bloch oscillations, resulting in collisionless bubbles with a time period $t_2 = 2\pi/\chi$.

### 3.2.2 Semiclassical quantisation of bubbles

Semiclassical quantisation of bubbles follows the procedure for the mesons, with the difference that the dynamical phase must be computed by integrating along the classical orbit connecting the turning points $k_a$ and $-k_a$ through $\pi$ and that the sign of $\chi$ is flipped.

Performing these changes in Eq. (41) and dividing by 2 leads to the following quantisation relation for bubbles in dynamical regime (I)

$$\mathcal{E}_n(K)(\pi - k_a) - \int_{k_a}^{\pi} dk\, \omega(k; K) = -\pi\chi\left(n - \frac{1}{2} + \frac{(-1)^{\kappa}}{4}\right), \tag{48}$$

together with the relation $\omega(k_a; K) = \mathcal{E}_n(K)$, and $\kappa = 0$ (1) corresponding to spatially odd (even) states as before. The notation $\mathcal{E}_n$, as opposed to $E_n$, is used to distinguish bubble excitations from the mesonic ones considered earlier.

A similar argument can be applied to the dynamical regime (III), but we omit the result since it is not needed in the sequel.

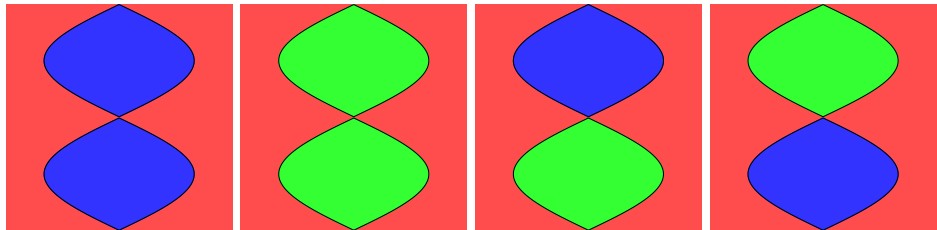

Figure 18: Pictorial representation of kink-antikink configurations forming neutral mesons (or bubbles) at rest ($K = 0$).

In the dynamical regime (II), the bubbles are collisionless, and the relevant equation can simply be obtained by changing the sign of $\chi$ in (43):

$$\mathcal{E}_n = -n\chi + \frac{1}{\pi} \int_{-\pi}^{\pi} dk\, \epsilon(k). \tag{49}$$

The energy difference between adjacent levels is again given by the Bloch angular frequency $\chi$.

Note that while the meson spectrum is bounded from below, the bubble spectrum is bounded from above, which corresponds to the metastable nature of the false vacuum state.

## 3.3 Including short-range interactions

Our calculations so far ignored that the kinks in the Potts quantum spin chain are interacting even when in the absence of longitudinal fields, i.e., in the pure transverse case (A.1). The relevant interaction is short-range, and the scattering theory was studied in detail in [40]. The field theory resulting in the scaling limit is integrable, and its scattering matrix is exactly known [55, 56]. However, the lattice case is not integrable therefore the scattering amplitudes must be determined numerically. In [40], it was found that the lattice phase shifts show some significant deviations from the exact field theory $S$-matrix, which were later found to be explained by contributions from irrelevant operators [57]. As a result, we determined the scattering amplitudes needed for our computations numerically using exact diagonalisation in finite volume, with the details given in Appendix A.3.

The effect of short-range interaction can be accounted for by adding the scattering phase shift to the integrated phase of the wave function

$$\oint dx\, k(x), \tag{50}$$

computed over a period of the motion. This is essentially the same procedure that was followed in the scaling limit [25], and the inclusion of this correction was found to be essential for an accurate description of the spectrum [26].

For neutral mesons in the dynamical regime (I), a pictorial representation of the relevant kink configurations is shown in Fig. 18, where the red domain is the true vacuum, and the internal domain can be either one of the false vacua (blue/green in the figure). The scattering process of the kinks must be diagonalised in the green/blue subspace, resulting in odd ('singlet') and even ('triplet') channels, which have different amplitudes that are determined in Appendix A.3.1. These internal configurations must be combined with a specific spatial parity, as reflected in the resulting modification of (41):

$$2E_n(K)k_a - \int_{-k_a}^{k_a} \omega(k; K) = \chi\left[2\pi(n - 1/2 + (-1)^\kappa/4) - \delta^{(\kappa)}(K/2 - k_a, K/2 + k_a)\right], \tag{51}$$

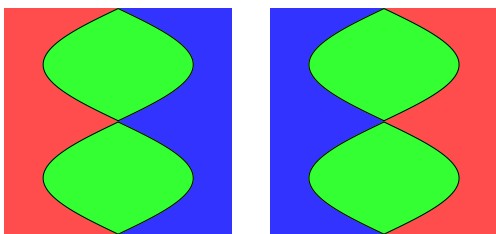

Figure 19: Pictorial representation of kink-kink configurations forming charged mesons (or bubbles) at rest ($K = 0$).

where

$$\delta^{(0)}(k_1, k_2) = \delta_s(k_1, k_2) \quad \text{and} \quad \delta^{(1)}(k_1, k_2) = \delta_t(k_1, k_2), \tag{52}$$

specify the matching between spatial and internal parities, which is shown to agree with the exact diagonalisation spectrum in Subsection B. As before, the turning point $k_a$ is determined from the condition $\omega(k_a; K) = E(K)$. For the neutral bubbles, the picture of configurations looks the same as in Fig. 18, but now the red domain is the false vacuum and the internal green/blue are the true vacua, and similar reasoning leads to the following modification of (48):

$$2\mathcal{E}_n(K)(\pi - k_a) - 2\int_{k_a}^{\pi} dk\, \omega(k; K) = -\chi\left[2\pi(n - 1/2 + (-1)^{\kappa}/4) - \delta^{(\kappa)}(K/2 + k_a, K/2 - k_a)\right]. \tag{53}$$

Here, there is a slight difference in the order of momenta inside the phase shift compared to Eq. (51), which corresponds to the middle point of the motion being $k = \pi$ instead of $k = 0$, and which is further justified by the agreement with the numerically determined spectrum as shown in Subsection B. This results in effectively swapping the sign in front of the phase shift since unitarity implies

$$\delta^{(\kappa)}(K/2 + k_a, K/2 - k_a) = -\delta^{(\kappa)}(K/2 - k_a, K/2 + k_a). \tag{54}$$

Turning now to the dynamical regime (II), which corresponds to collisionless states for mesons and bubbles, no short-range correction is needed due to the absence of scattering, so quantisation conditions (43) and (49) survive unmodified.

## 3.4 Charged mesons and charged bubbles

When the true vacuum is doubly degenerate, it is possible to have charged mesons for which the internal part is the false vacuum, while the left/right external true vacuum domains are different. This is depicted in Fig. 19, where the internal green part is the false vacuum and the true vacua are red and blue. This is different from the kinks associated with the doubly degenerate true vacuum states since the latter directly interpolate between the true vacua. In contrast, the charged meson configurations travel through the false vacuum, as shown in Fig. 19. The scattering is diagonal in this case, with the corresponding kink-kink phase shift determined in A.3.2. Using the same arguments as for the neutral mesons, the relevant quantisation relation is

$$2E_n(K)k_a - \int_{-k_a}^{k_a} dk\, \omega(k; K) = \chi[2\pi(n - 1/4) - \hat{\delta}(K/2 - k_a, K/2 + k_a)]. \tag{55}$$

In the case of doubly degenerate false vacua, Fig. 19 can be reinterpreted as the depiction of a charged bubble, where the internal green part is the true vacuum and the false vacua are red and blue. The relevant quantisation relation then takes the form

$$2\mathcal{E}_n(K)(\pi - k_a) - 2\int_{k_a}^{\pi} dk\,\omega(k;K) = -\chi[2\pi(n - 1/4) + \hat{\delta}(K/2 - k_a, K/2 + k_a)]. \qquad (56)$$

## 4 Theory vs. phenomenology

In this Section, we show how our understanding of the spectrum can be applied to explain the phenomenology of the different quenches described in Section 2.

### 4.1 Quench spectroscopy

The spectral analysis of the time evolution of the magnetisation provides detailed information about the excitation spectrum of the system. This quench spectroscopy proved to be very useful in analysing the dynamics of real-time confinement [4], as well as anticonfinement in the Ising chain [22]. Following these works, we analyse the Fourier spectrum of the time evolution of the magnetisation(s) $M_\mu$ after the corresponding quenches.

The time evolution is simulated using the iTEBD method with second-order Trotterisation. The time step was set to $\delta t = 0.005$, and we performed simulations with maximal bond dimensions $\chi_{\max} = 300$ and $\chi_{\max} = 800$; however, we only include the results with $\chi_{\max} = 300$ in the main text. The results with $\chi_{\max} = 800$, together with the other details of the numerics, can be found in Appendix C. The definition of the Fourier transform we apply is given by

$$M(\omega) \equiv M\left(\frac{2\pi k}{N\delta t}\right) := \frac{1}{\sqrt{N}}\sum_{n=0}^{N-1} e^{-2\pi i \frac{kn}{N}} M(n\delta t), \qquad (57)$$

where $N$ is the total number of the discrete time steps of the simulation, and $k$ is an integer such that $k \in [0, \dots, N-1]$.

The resulting Fourier spectra are compared to quasiparticle mass gaps calculated by exact diagonalisation (ED) with periodic boundary conditions, typically with chains of length $L = 10$. To disentangle 2 and 3-kink bound states in the ED spectrum, we used the operator (A.6) as discussed in Appendix A.2. The identification of the different species of mesons and bubbles, as well as the kink-antikink states of effective Ising subsystems, was based upon the matching with the semiclassical predictions obtained in Section 3. Details of this identification are discussed in Section B where we also calculate the energy levels of the effective Ising subsystems.

#### 4.1.1 Positively aligned: Standard confinement

In the presence of a positive $h_1$ field, there is a single true vacuum and doubly degenerate false vacuum states. The initial state (10) is favoured by the longitudinal field. Therefore, this quench is dominated by real-time confinement, and the contributing excitations are expected to be quasiparticles built upon the true vacuum, predominantly mesons, analogously to the Ising case considered in [4]. These mesonic states form a sequence indexed by a species number $n$, and their overlaps with the initial state are expected to decrease with $n$. Additionally, since the initial state (10) is even under the unbroken charge conjugation $\mathcal{C}$ exchanging the colours 2 and 3, only $\mathcal{C}$-even mesons are expected to contribute. Furthermore, the translation invariance of the initial state dictates that the contributing states all have zero total momentum $K = 0$.

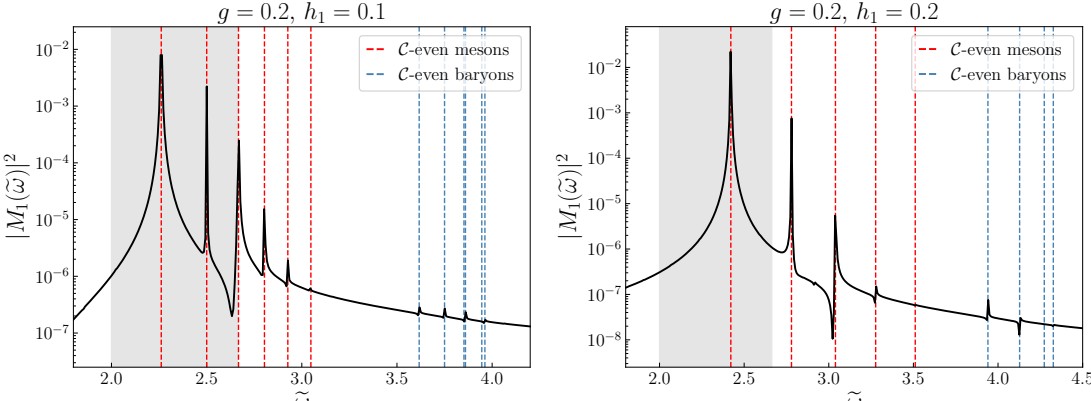

Figure 20: The Fourier spectrum of $M_1$ in terms of $\widetilde{\omega} = \omega/m_k$ in the standard confining case for the parameters $g = 0.2$, $h_1 = 0.1$ (left) and $h_1 = 0.2$ (right). The dashed red lines correspond to the respective low energy $\mathcal{C}$-even meson masses, while the dashed blue lines denote the low energy $\mathcal{C}$-even baryon masses. The masses were calculated via ED with $L = 10$ and PBC as shown in Fig. 34 and they match very well with the position of the peaks. The background is coloured grey in the interval $(\tilde{\omega}_{\min}, \tilde{\omega}_{\max}) = (2, 2.663)$; collisional/collisionless mesons lie inside/outside this interval, respectively.

In this case, there is only a single independent magnetisation, which can be chosen as $M_1(t)$, for which the typical Fourier spectrum is illustrated by the cases shown in Fig. 20. The frequency variable is normalised by the pure transverse kink mass as $\widetilde{\omega} = \omega/m_k$ where $m_k$ is given by (A.4). The first few $\mathcal{C}$-even meson masses

$$m_n = E_n(K = 0),\tag{58}$$

calculated via ED with $L = 10$ and PBC as shown in Fig. 34, are shown by red dashed lines in Fig. 20, scaled by the kink mass $m_k$, and they match the dominant peaks with high precision. Within the semiclassical framework, the energies of the collisional mesons (described by regime (I) in Subsection 3.1) in $m_k$ units lie in the interval $(\tilde{\omega}_{\min}, \tilde{\omega}_{\max})$ given by the two-kink continuum spectrum in the pure transverse chain (A.1) in the thermodynamic limit $L \to \infty$. From the pure transverse kink dispersion relation (A.3) we get $\tilde{\omega}_{\min} = 2$ and $\tilde{\omega}_{\max} = 2\sqrt{A-B}/\sqrt{A+B}$, which can be computed from the data in Table 1. In Fig. 20, we indicate the interval $(\tilde{\omega}_{\min}, \tilde{\omega}_{\max})$ by colouring the background grey to help the readers distinguish collisional and collisionless mesons which lie inside and outside this interval respectively.

Interestingly, we also observe peaks of higher energy in the Fourier spectrum. These peaks can be explained by $\mathcal{C}$-even baryon excitations, which can be identified as the lowest energy states in the 3-kink part of the spectrum, that can be separated from the 2-kink spectrum using the operator (A.6). The $\mathcal{C}$-even baryon masses calculated via ED are shown by blue dashed lines in Fig. 20, and they indeed match the position of these higher peaks very well.

We finally comment that depending on the volume, there can be meson states that are mixed among the baryon levels in the ED spectrum. However, the position of these highly excited meson levels depends on the volume, while that of the lowest-lying baryon levels is essentially independent of the volume, which makes their identification unambiguous. Additionally, while in infinite volume the meson spectrum is unbounded from above, their overlaps with the initial state decrease strongly with their energy (which is also apparent from Fig. 20), and those in the energy range of the baryon peaks are expected to have too small overlaps to show up in the quench spectrum.

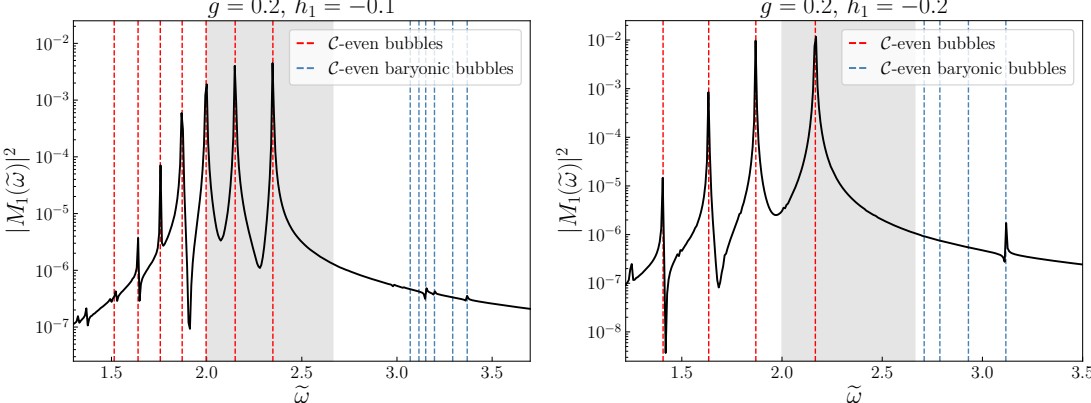

Figure 21: The Fourier spectrum of $M_1$ in terms of $\widetilde{\omega} = \omega/m_k$ in the standard anti-confining case for the parameters $g = 0.2$, $h_1 = -0.1$ (left) and $h_1 = -0.2$ (right). The dashed red lines correspond to the respective first few $\mathcal{C}$-even bubble masses, while the dashed blue lines denote the first few $\mathcal{C}$-even baryonic bubble masses. The masses were calculated via ED with PBC and $L = 10$ and $L = 8$ respectively, relative to the energy of the false vacuum as shown in Fig. 38 and they match very well with the position of the peaks. The background is coloured grey in the interval $(\tilde{\omega}_{\min}, \tilde{\omega}_{\max}) = (2, 2.663)$; collisional/collisionless bubbles lie inside/outside this interval, respectively.

### 4.1.2 Negatively aligned: Standard anticonfinement

Turning on a negative $h_1$ results in a doubly degenerate true and in a single false vacuum state. The initial state (10) is then disfavoured by the longitudinal field, leading to an anticonfining quench whose Ising analogue was examined in [22]. Since the initial state is aligned with the false vacuum, the relevant excitations are expected to be bubbles built upon the false vacuum. Once again, the contributing states must be $\mathcal{C}$-even with total momentum $K = 0$. The bubble spectrum is unbounded from below, and the energy decreases with increasing bubble size, i.e., with their species label. The spectrum is expected to be dominated by the highest energy bubbles since their overlap with the false vacuum decreases steeply with their size (corresponding to the volume of true vacuum nucleated inside the initial false vacuum state) [23]. Note that while meson overlaps decrease with energy, the bubble overlaps show the opposite behaviour, which can be used to distinguish a false from a true vacuum as suggested in [49].

Two typical examples for the Fourier spectrum of $M_1(t)$ are shown in Fig. 21 with bubble energies (computed relative to the false vacuum) indicated by dashed red lines. The bubble masses were calculated via ED with PBC and $L = 10$ and $L = 8$ respectively, relative to the energy of the false vacuum as shown in Fig. 38. Again, the first few most dominant peaks match the ED bubble masses with high precision, and the overlaps indeed depend on energy the opposite way compared to the case of mesons in confining quenches. As in the case of the positively aligned quench, the background of Fig. 21 is coloured grey in the interval $(\tilde{\omega}_{\min}, \tilde{\omega}_{\max})$, and the collisional/collisionless bubbles lie inside/outside this interval, respectively.

We also find peaks at 3-kink states built on top of the false vacuum, which we call baryonic bubbles, with their ED masses shown by blue dashed lines in Fig. 21. Like in the (two-kink) bubble case, we expect the baryonic bubbles with the highest energy to contribute the most, which is indeed the case.

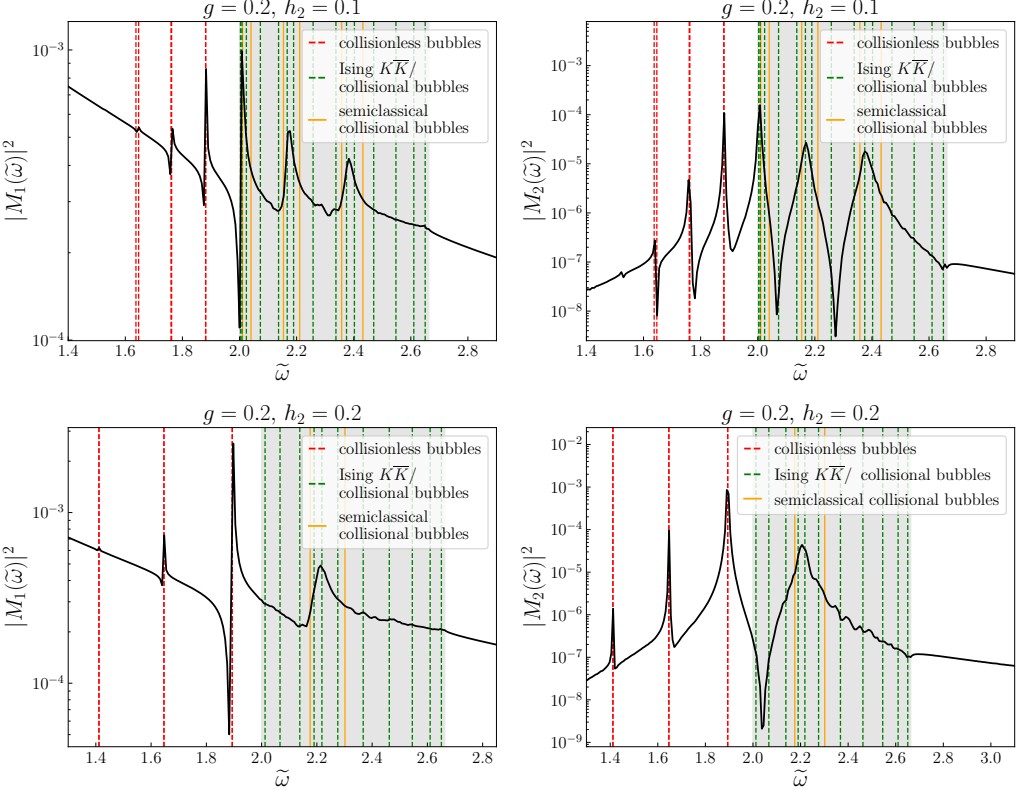

Figure 22: The Fourier spectrum of $M_1$ (left) and $M_2$ (right) in terms of $\widetilde{\omega} = \omega/m_k$ in the partial anticonfining case for the parameters $g = 0.2$, $h_2 = 0.1$ (top) and $h_2 = 0.2$ (bottom). The dashed lines denote the high-energy (both $\mathcal{C}$-even and $\mathcal{C}$-odd) 2-kink masses. The green lines correspond to the hybridised Ising $K\overline{K}$ and collisional bubble states, while the red ones to the collisionless bubbles. The red lines match very well with the sharp peaks, while the top edge of the two-kink continuum (the highest energy green line) corresponds to a small cusp in the quench spectrum. The positions of the wide resonant peaks are indeed in the vicinity of the semiclassical collisional bubble masses denoted by solid orange lines. The background is coloured grey in the interval $(\widetilde{\omega}_{\min}, \widetilde{\omega}_{\max}) = (2, 2.663)$ where the collisional bubbles lie. All masses were calculated via ED with PBC and $L = 10$ relative to the energy of the false vacuum as shown in Fig. 34.

### 4.1.3 Positive oblique: Partial anticonfinement

Turning on a positive $h_2$, the initial state (10) is polarised in the direction of one of the two false vacua. As a result, the quench is anticonfining, and the dominant contribution is expected to come from bubble states built on top of the false vacua.

The initial state (10) can be written as

$$|\psi_0\rangle = |111\cdots1\rangle = \frac{1}{\sqrt{2}}\{|\mathcal{C}\text{-even}\rangle + |\mathcal{C}\text{-odd}\rangle\}, \tag{59}$$

where

$$|\mathcal{C}\text{-even}\rangle = \frac{1}{\sqrt{2}}\{|111\cdots1\rangle + |333\cdots3\rangle\}, \quad |\mathcal{C}\text{-odd}\rangle = \frac{1}{\sqrt{2}}\{|111\cdots1\rangle - |333\cdots3\rangle\}, \tag{60}$$

are $\mathcal{C}$-even and $\mathcal{C}$-odd states respectively under $\mathcal{C}$ exchanging 1 and 3. As a result, both even and odd quasiparticles can contribute. To match the peaks of the Fourier spectrum, the excitation energies must be computed relative to the false vacuum state.

We show typical Fourier spectra of $M_1(t)$ and $M_2(t)$ for these quenches in Fig. 22. Note that in addition to the bubble states, we expect a contribution from the effective Ising subsystem of kinks interpolating between the two false vacua. In the ED spectrum (Fig. 34), these excitations can be found in the top part of the 2-kink levels. However, in infinite volume, the two-particle states form a continuum. Therefore, we do not expect peaks at the discrete locations of the corresponding zero-momentum ED eigenstates calculated in finite volume $L$. Rather, a threshold in the Fourier spectrum must appear where the effective Ising two-kink continuum ends. Indeed, such a threshold is visible at the top edge of the two-kink levels, manifested as a small cusp in the spectrum (see also in Fig. 34).

The presence of these kink excitations is responsible for the partial nature of anticonfinement (a.k.a. Wannier-Stark localisation), manifested in the growth of entanglement entropy shown in Fig. 10b, as well as in the presence of unsuppressed light-cone behaviour in certain two-point correlations (Fig. 11), which is discussed in detail in Subsection 4.2 below.

In Fig. 22 we represent high-energy 2-kink masses by dashed lines. The masses were calculated via ED with PBC and $L = 10$ relative to the energy of the false vacuum as shown in Fig. 34. Since the matrix element $s_2(k_1, k_2)$ of the S-matrix given by equation (A.10) is non-zero, the collisional bubbles lying in the effective Ising two-kink continuum cannot exist in the infinite chain as stable excitations. Instead, they hybridise with the continuum of effective Ising two-kink states. As explained in the standard quench scenarios, the interval $(\tilde{\omega}_{\min}, \tilde{\omega}_{\max}) = (2, 2.663)$ containing the collisional bubbles is again coloured grey. It turns out that all the ED masses that lie inside the interval $(\tilde{\omega}_{\min}, \tilde{\omega}_{\max})$ also fall into the energy band determined by the effective Ising two-kink states that can be calculated using (B.4). As a result, all the collisional bubbles are hybridised with the effective Ising two-kink states. We denote all the ED masses that lie inside the interval of the effective Ising two-kinks by green dashed lines representing the hybridised states.

However, collisional bubbles can still exist as unstable particles (resonances) on the infinite chain. In the Fourier spectra of the post-quench oscillations, shown in Fig. 22, these unstable collisional bubbles result in wide resonant peaks expected to be in the vicinity of semiclassically calculated masses of the collisional bubbles, which are shown by orange solid lines in the figure. In contrast, at the position of the masses of the collisionless bubbles (represented by red dashed lines), we expect sharp peaks. Fig. 22 confirms a good agreement between the theoretical expectations and the quench spectroscopy results. We also note that at the present resolution, we could not identify any baryonic (i.e. 3-kink) excitations in the spectrum.

### 4.1.4 Negative oblique: Partial confinement

In the presence of a negative $h_2$, the initial state (10) which can be again written as a combination of $\mathcal{C}$-even and $\mathcal{C}$-odd states as (59), is favoured by the longitudinal field and it has finite overlaps with one of the true vacua leading to a (partially) confining quench. Therefore, the dominant contribution to quench spectroscopy is expected to come from mesonic excitations, both $\mathcal{C}$-even and $\mathcal{C}$-odd. The presence of the Ising subsystem built upon the doubly degenerate true vacua leads to effective Ising kink-antikink states, which now occupy the bottom part of the low-energy spectrum as visible in Fig. 38. They are responsible for the partial nature of confinement, manifested in the growth of entanglement entropy shown in Fig. 12b, as well as in the presence of unsuppressed light-cone behaviour in certain two-point correlations (Fig. 13), which is discussed in detail in Subsection 4.2 below.

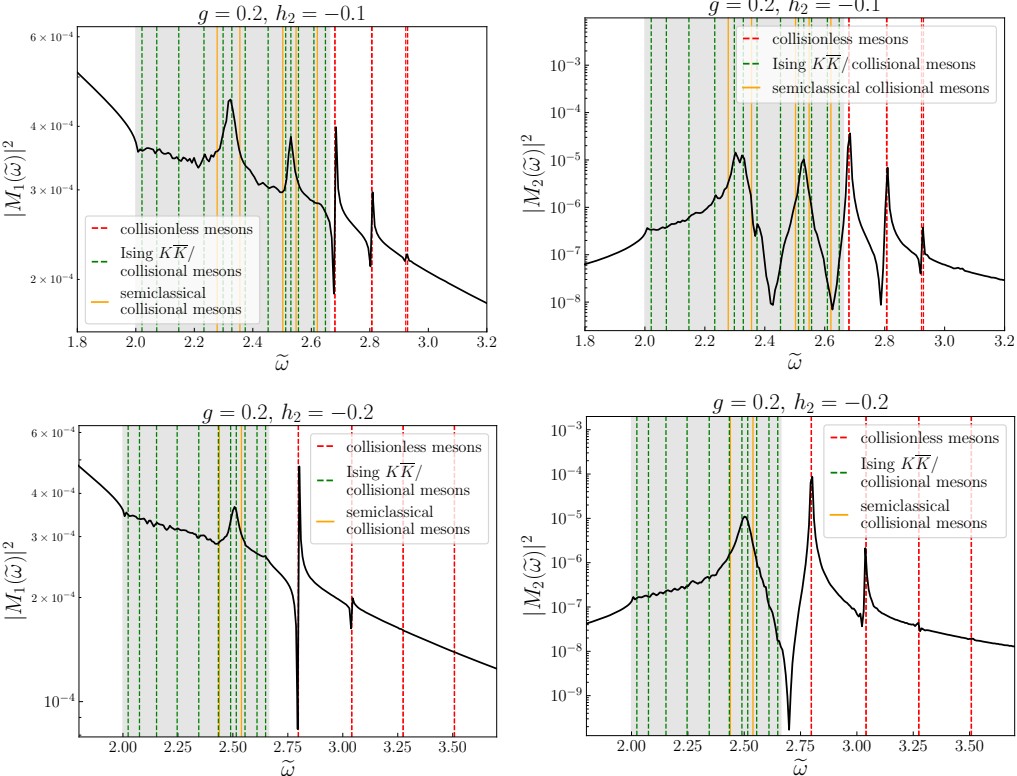

Figure 23: The Fourier spectrum of $M_1$ (left) and $M_2$ (right) in terms of $\widetilde{\omega} = \omega/m_k$ in the partial confining case for the parameters $g = 0.2$, $h_2 = -0.1$ (top) and $h_2 = -0.2$ (bottom). The dashed lines denote the low-energy (both $\mathcal{C}$-even and $\mathcal{C}$-odd) 2-kink masses. The green lines correspond to the hybridised Ising $K\bar{K}$ and collisional meson levels, while the red ones show the collisionless mesons. The red lines match very well with the sharp peaks, while the threshold of the two-kink continuum (the lowest energy green line) corresponds to a small cusp in the quench spectrum. The positions of the wide resonant peaks are indeed in the vicinity of the semiclassical collisional meson masses denoted by solid orange lines. The background is coloured grey in the interval $(\tilde{\omega}_{\min}, \tilde{\omega}_{\max}) = (2, 2.663)$ where the collisional mesons lie. All masses were calculated via ED with PBC and with $L = 10$ as shown in Fig. 38.

Typical Fourier spectra of $M_1(t)$ and $M_2(t)$ for these quenches can be seen in Fig. 23. The dashed lines in Fig. 23 correspond to low-energy 2-kink ED masses that were calculated via ED with PBC and with $L = 10$ as shown in Fig. 38. The presence of the effective Ising two-kink continuum again leads to a cusp signalling the threshold of the corresponding two-particle continuum in the infinite volume system. The collisional meson masses lie in the interval $(\tilde{\omega}_{\min}, \tilde{\omega}_{\max})$, as explained in Subsection 4.1.1, which is indicated by the background coloured grey. Similarly to the previous case, the collisional mesons hybridise with the effective Ising two-kink states. The ED masses corresponding to the hybridised states are denoted by dashed green lines. Once again, the collisional mesons lead to wide resonant peaks which are located in the vicinity of the semiclassical collisional meson masses (shown by orange solid lines). The red dashed lines denote the collisionless meson masses obtained from ED.

The agreement between the ED spectrum and quench spectroscopy is again convincing. Furthermore, similar to the previous case, there is no sign of 3-kink bound states (i.e., baryons) at the present resolution.

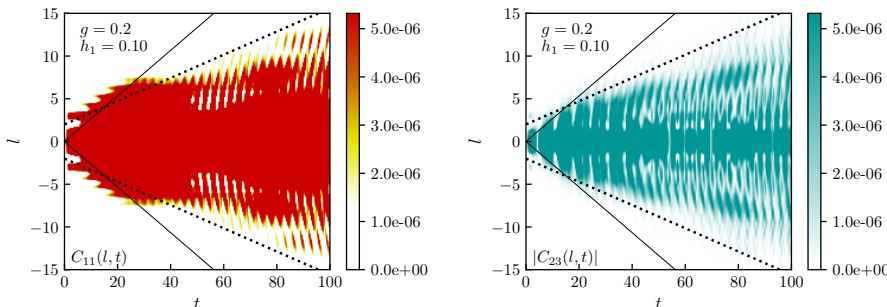

Figure 24: Connected correlators $C_{11}$ (*left*) and $C_{23}$ (*right*) in the positively aligned (confining) quench scenario. The colour scale was changed relative to Fig. 7 to emphasize the escaping fronts present despite their suppression by the confining force. The black solid lines are drawn at $\pm 2v_{LR}t$ with $v_{LR}$ being the maximum velocity of the kinks in the pure transverse model given by (A.5). In contrast, the dotted black lines are determined by the maximum meson velocity computed from the semiclassical dispersion relations.

## 4.2 Localisation and light-cone spreading in two-point correlations

### 4.2.1 Aligned quenches

For positively aligned quenches, the dynamics is determined by real-time confinement of the kink excitations. As in [4], this leads to a strong suppression of the light-cone structure of the connected correlation functions as shown in Fig. 7. Nevertheless, as already observed in the case of the Ising model [4], subleading effects lead to the presence of a light cone traced out by two-meson states, which is suppressed by several orders of magnitude, as shown in Fig. 24, with its slope determined by the maximum velocity of mesonic excitations. The largest maximum velocity is obtained for the lightest meson corresponding to $n = 1$ in (51), and can be computed as

$$v_{\mathrm{M}} = \max_K \frac{dE_1(K)}{dK}. \tag{61}$$

In Figs. 7 and 24, the black dotted lines are drawn at $\pm 2v_{\mathrm{M}}t$.

For negatively aligned (anticonfining) quenches, the spreading of correlations is suppressed by Bloch oscillations [22], giving rise to Wannier-Stark localisation [6,10]. This again leads to a strong suppression of the light-cone structure of the connected correlation functions, as shown in Fig. 9. Nevertheless, similarly to the positively aligned case, subleading effects corresponding to the emission of bubble pairs lead to the presence of a light cone, which is suppressed by several orders of magnitude, as shown in Fig. 25. The main difference from the confining case is that the slope of this light cone is determined by the maximum velocity of bubble excitations. In this case, the fastest excitation is the heaviest bubble corresponding to species number 1 in the semiclassical quantisation (53). Since the initial state is aligned with the false vacuum, this is the state which dominates the dynamics, as also evident from the quench spectroscopy results shown in Fig. 21. Therefore, in Figs. 9 and 25, the black dotted lines are drawn at $\pm 2v_{\mathrm{B}}t$ with

$$v_{\mathrm{B}} = \max_K \frac{d\mathcal{E}_1(K)}{dK}. \tag{62}$$

We remark that in both Figs. 24 and 25, some residual correlations can be observed outside the light cone indicated by the lines. The reason is that the light cone lines are drawn from a single point, while initial state correlations have a non-zero, albeit very short, range.

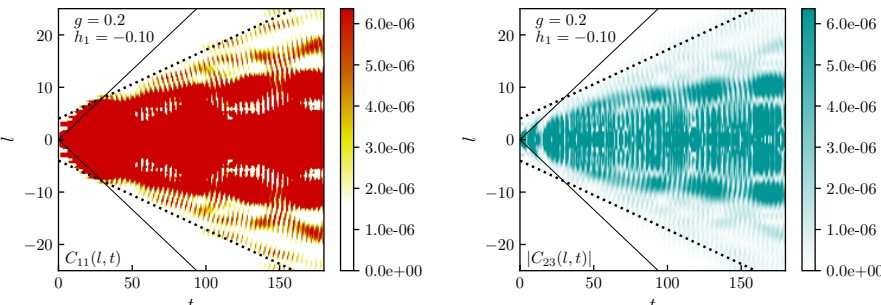

Figure 25: Connected correlators $C_{11}$ (*left*) and $C_{23}$ (*right*) in the negatively aligned (anticonfining) quench scenario. The colour scale was changed relative to Fig. 9 to emphasize the escaping fronts present despite their suppression by the confining force. The black solid lines are drawn at $\pm 2 v_{LR} t$ with $v_{LR}$ being the maximum velocity of the kinks in the pure transverse model given by (A.5). In contrast, the black dotted lines are determined by the maximum bubble velocity computed from the semiclassical dispersion relations.

### 4.2.2 Oblique quenches

In the oblique quenches, the longitudinal field points in direction 2, which results in the degeneracy of the vacua 1 and 3, leading to an effective Ising subsystem that is excited by the initial state polarised along direction 1. As a result, connected correlations $C_{11}$ propagate unsuppressed, with the kink velocity determined from (B.2) as

$$v_{\mathrm{I}} = \max_k \frac{d\epsilon_{\mathrm{Ising}}(k)}{dk} = h_{\mathrm{eff}} = \frac{2g}{3}\,. \tag{63}$$

The corresponding light cones appear in both positive and negative oblique quenches, shown in Figs. 11a and 13a, respectively, where the dashed lines correspond to $\pm 2 v_{\mathrm{I}} t$.

The correlation $C_{22}$ shows the full effect of anticonfinement/confinement (depending on the specific case), as discussed for negatively/positively aligned quenches. In the positive oblique case shown in Fig. 11, the spreading of these correlations is suppressed by Wannier-Stark localisation, while in the negative oblique case in Fig. 13, the relevant mechanism is real-time confinement. This is also confirmed by the appearance of bubble/meson excitations in their quench spectroscopy as discussed in Subsections 4.1.3 and 4.1.4. Additionally, the excitation of the effective Ising kink system results in light-cone propagation in the (partially suppressed) $C_{23}$ correlator. This is demonstrated in Figs. 11c and 13c, where the dashed lines correspond to the effective Ising kink velocity (63).

To sum up, in oblique quenches, the light-cone spreading of correlations is only partially suppressed, further justifying the terminology of partial (anti)confinement used above.

## 5 Conclusions

In the present work, we investigate non-equilibrium dynamics after quantum quenches on the mixed-field three-state Potts quantum chain in the ferromagnetic regime. The 3-state Potts model is a natural generalisation of the Ising model, where the local degrees of freedom have three possible orientations instead of two. As a result, the phenomenology of the Potts model is much richer. In both cases, switching on a (weak) longitudinal field in the ferromagnetic

phase leads to confinement, which can be considered an analogue model of quantum chromodynamics. However, in the Ising model, the confined excitations can only be two-kink bound states analogous to mesons, while the Potts model also allows for three-kink bound states, which are the analogues of baryons. In this connection, we note that the realisation of baryonic excitations in condensed matter systems has attracted recent interest [58–60].

Beyond the spectrum, the non-equilibrium dynamics of the Potts model in the ferromagnetic phase is also more varied than that of the Ising case. In the latter, there are essentially two scenarios. The case when the longitudinal field is parallel to the initial magnetisation leads to dynamical confinement [4], while in the antiparallel (anticonfining) case the time evolution corresponds to the decay of the false vacuum [18, 52], which however differs from the usual scenario [20] since the nucleated true vacuum bubbles are prevented from expanding by Bloch oscillations [22]. As a result, both cases show suppression of both the growth of entanglement entropy and the spreading of correlations, which are qualitatively similar despite the difference between the underlying mechanisms.

In the Potts model, quenches with the longitudinal field aligned with the initial magnetisation are essentially analogous to the Ising case. Nevertheless, there are some interesting modifications. For the positively aligned case, we found dynamical confinement as in the Ising case [4], with two alterations: firstly, the presence of baryonic excitations, which was verified using the quench spectroscopy. Secondly, although the time evolution of the entanglement entropy is dominated by oscillations, as in the Ising case, there is an additional long-time drift, which can be attributed to the presence of (albeit suppressed) escaping fronts. In the negatively aligned case, we found Wannier-Stark localisation caused by Bloch oscillations; however, due to the degeneracy of the true vacuum states, the nucleated bubbles can now also be three-kink states, i.e., of a baryonic nature. In addition, the entanglement entropy again shows a slow drift for the same reason as in the confining case.

For the oblique quenches, when the longitudinal field is not aligned with the initial magnetisation, there is no parallel with the Ising case. In such cases, both confinement and Wannier-Stark localisation are only partially effective, leading to the unsuppressed growth of entanglement entropy and the presence of substantial light-cone spreading of certain correlations. The reason is that the presence of three ground states leads to degeneracies for either the true or the false vacuum states depending on the sign of the longitudinal field, resulting in an unconfined effective Ising kink subsystem. This implies that in the case of oblique quenches, where the longitudinal field is not parallel to the initial magnetisation, the localisation effects can only be partial, while certain correlations spread unsuppressed. This also results together in an unsuppressed growth of entanglement entropy.

There are several interesting open problems. First, developing a description for the baryon spectrum similar to that of the mesons and bubbles. This requires a quantum mechanical description of the three-body problem, which generalises the existing field-theoretic approach [27] to the lattice model. Second, it seems interesting to explore transport properties and local quench protocols, following similar work done for Ising quenches [6, 23]. Finally, it would be interesting to see whether the various phenomena explored here can be realised in experimental settings, based e.g. on recent proposals to implement the Potts universality class in Rydberg atom systems [61–63].

# Acknowledgments

We are grateful to M.A. Werner for sharing his extensive knowledge of MPS methods which proved very helpful in developing our simulations.



**Funding information**  GT was partially supported by the Ministry of Culture and Innovation and the National Research, Development and Innovation Office (NKFIH) through the OTKA Grant K 138606 and also under the Quantum Information National Laboratory of Hungary (Grant No. 2022-2.1.1-NL-2022-00004). AK was also partially supported by the Doctoral Excellence Fellowship Programme (DCEP), funded by the National Research Development and Innovation Fund of the Ministry of Culture and Innovation and the Budapest University of Technology and Economics, under a grant agreement with the National Research, Development and Innovation Office. OP acknowledges support from ERC under Consolidator Grant number 771536 (NEMO). This collaboration was partially supported by the CNR/MTA Italy-Hungary 2023-2025 Joint Project "Effects of strong correlations in interacting many-body systems and quantum circuits". We also acknowledge KIFÜ (Governmental Agency for IT Development, Hungary) for awarding us access to the Komondor HPC facility based in Hungary.

# A  Spectrum and scattering in the pure transverse chain

## A.1  Dispersion relation

In the ferromagnetic phase of the pure transverse Potts chain

$$H = -\sum_{i=1}^{L}\sum_{\mu=1}^{3}(P_i^{\mu}P_{i+1}^{\mu}) - g\sum_{i=1}^{L}\tilde{P}_i \,, \tag{A.1}$$

the quasiparticles are kinks/antikinks. In finite volume, single-kink states obey twisted boundary conditions

$$P_{L+1}^{\mu} = P_{L+1}^{\mu\pm 1} \,, \tag{A.2}$$

where $\mu\pm 1$ is understood mod 3, and the upper/lower signs correspond to the kink/anti-kink excitations. Using exact diagonalisation with a finite $L$, the energy of a single-kink excitation of a given momentum $k$ can be determined. Note that the momentum $k$ is quantised in units of $2\pi/3L$ due to the twisted boundary conditions being only periodic under a shift of length $3L$. The dispersion relation can then be obtained by subtracting the ground state of the chain with the same length $L$ and periodic boundary conditions, with the results illustrated in Fig. 26. Following the Ising case, it can be fitted with a function of the form (see Fig. 26)

$$\epsilon(k) = \sqrt{A + B\cos k} \,. \tag{A.3}$$

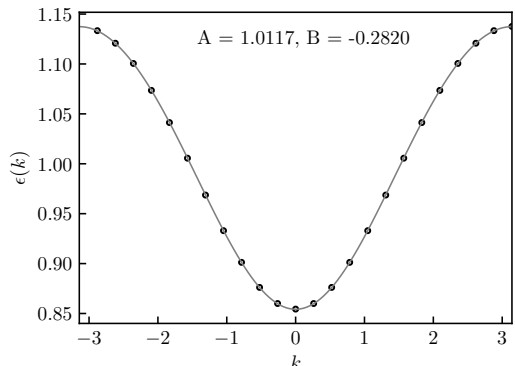

Figure 26: Dispersion relation for the kink excitation obtained with exact diagonalisation of a chain of length $L = 8$ with twisted boundary conditions and transverse field $g = 0.2$. The dots represent numerical data, while the solid line is the fit with the function $\sqrt{A + B\cos k}$.

Table 1: Values of $A$ and $B$ of the dispersion relation $\epsilon(k) = \sqrt{A + B \cos k}$ for different values of the transverse field $g$.

|   | $g = 0.2$ | $g = 0.3$ | $g = 0.4$ | $g = 0.5$ | $g = 0.6$ |
|---|---|---|---|---|---|
| $A$ | 1.0117 | 1.0291 | 1.0565 | 1.0955 | 1.1479 |
| $B$ | $-0.2820$ | $-0.4321$ | $-0.5863$ | $-0.7434$ | $-0.9023$ |

Values of $A$ and $B$ are presented in Table 1 for different coupling values $g$. Furthermore, we define the kink mass as

$$m_k = \epsilon(0) = \sqrt{A + B}\,, \tag{A.4}$$

and the Lieb-Robinson velocity of kinks as

$$v_{LR} = \max_k \frac{d\epsilon(k)}{dk}\,. \tag{A.5}$$

## A.2 Identifying kink-antikink states

In the neutral sector, selecting kink-antikink states is often difficult, as they overlap with three-kink states of zero total charge. We note that these are effectively the same states that become baryonic excitations once a confining longitudinal field is switched on, making their identification important as well. To accomplish this selection, it is necessary to have an operator whose expectation values correspond to the total number of kinks and antikinks (counted while neglecting their charge).

The following operator

$$\mathcal{O}_{\text{POTTS}} = \sum_i \left( P_i^1 P_{i+1}^2 + P_i^2 P_{i+1}^1 + P_i^2 P_{i+1}^3 + P_i^3 P_{i+1}^2 + P_i^3 P_{i+1}^1 + P_i^1 P_{i+1}^3 + \frac{2}{3} \right), \tag{A.6}$$

counts the number of pairs of adjacent sites that correspond to different colours. As a result, the expectation value of this operator returns approximately the number of kinks forming the state. However, it must be assumed that the typical extension of domain walls is very short (less than two sites), which is only valid for suitably small transverse fields when the kinks are sufficiently localised. Additionally, the longitudinal field must also be small enough so that the typical distance between confined kinks is more than two lattice sites. For the parameters we use in our exact diagonalisation studies, we found that these conditions are suitably satisfied, so the expectation value of operator (A.6) could reliably distinguish two-kink states from three-kink states.

## A.3 Kink scattering amplitudes

The scattering of two quasiparticles on each other can be characterised by the two-particle S-matrix, which relates the amplitude of an incoming asymptotic wave function

$$\psi_{k_1 \sigma_1, k_2 \sigma_2}(x_1 \ll x_2) \sim A^{\text{in}}_{\sigma_1, \sigma_2}(k_1, k_2) e^{i(k_1 x_1 + k_2 x_2)}\,, \tag{A.7}$$

with momenta $k_1 > k_2$, to that of the outgoing wave function

$$\psi_{k_1 \sigma_1, k_2 \sigma_2}(x_1 \gg x_2) \sim B^{\text{out}}_{\sigma_1, \sigma_2}(k_1, k_2) e^{i(k_1 x_1 + k_2 x_2)}\,, \tag{A.8}$$

as

$$\mathbf{B}^{\text{out}} = S(k_1, k_2)\mathbf{A}^{\text{in}}. \tag{A.9}$$

Imposing the $\mathbb{S}_3$ symmetry of the Potts model, the structure of the two-body S-matrix is further restricted to the form [40]:

$$S(k_1, k_2) = \begin{pmatrix} s_3(k_1, k_2) & 0 & 0 & 0 \\ 0 & s_1(k_1, k_2) & s_2(k_1, k_2) & 0 \\ 0 & s_2(k_1, k_2) & s_1(k_1, k_2) & 0 \\ 0 & 0 & 0 & s_3(k_1, k_2) \end{pmatrix}. \tag{A.10}$$

Here $s_3(k_1, k_2)$ describes the interaction in the charged sector consisting of kink-kink and antikink-antikink states:

$$K_{\alpha\gamma}(k_1)K_{\gamma\beta}(k_2) = s_3(k_1, k_2)K_{\alpha\gamma}(k_2)K_{\gamma\beta}(k_1), \qquad \alpha \neq \beta, \tag{A.11}$$

while $s_1(k_1, k_2)$ and $s_2(k_1, k_2)$ describe interactions in the neutral sector of kink-antikink states:

$$K_{\alpha\gamma}(k_1)K_{\gamma\alpha}(k_2) = s_1(k_1, k_2)K_{\alpha\gamma}(k_2)K_{\gamma\alpha}(k_1) + s_2(k_1, k_2)K_{\alpha\beta}(k_2)K_{\beta\alpha}(k_1), \qquad \beta \neq \gamma. \tag{A.12}$$

### A.3.1 Kink-antikink scattering

In the kink-antikink sector, the eigenvalues of the S-matrix read

$$\begin{aligned}
s_t(k_1, k_2) &= e^{i\delta_t(k_1, k_2)} = s_1(k_1, k_2) + s_2(k_1, k_2), \\
s_s(k_1, k_2) &= e^{i\delta_s(k_1, k_2)} = s_1(k_1, k_2) - s_2(k_1, k_2),
\end{aligned} \tag{A.13}$$

where we introduced the "triplet" and "singlet" eigenvalues $s_t(k_1, k_2)$ and $s_s(k_1, k_2)$ and the corresponding phase shifts $\delta_t(k_1, k_2)$ and $\delta_s(k_1, k_2)$. The two sectors correspond to the even and the odd sector respectively according to the parity of the $\mathbb{Z}_2$ charge conjugation symmetry generated by $\mathcal{C}$ introduced in (3). We compute the phase shifts from exact diagonalisation data on a finite chain of length $L$. For the kink-antikink we can both use periodic boundary conditions and partial twisted boundary conditions for which the end of the chain is twisted by the operator

$$\tilde{\mathcal{T}}_L = \mathbb{I}_1 \mathbb{I}_2 \cdots \tilde{T}_L, \tag{A.14}$$

with the operator $\tilde{T}_L$ being

$$\tilde{T}_L = \begin{pmatrix} 1 & 0 & 0 \\ 0 & 0 & 1 \\ 0 & 1 & 0 \end{pmatrix}, \tag{A.15}$$

acting on site $L$ exchanging colours 2 and 3.

The spectrum of kink-antikink states, selected by the operator (A.6), in a finite chain with partially twisted boundary conditions is reported in Fig. 27, where odd and even sectors of the restricted $\mathbb{Z}_2$ symmetry under the exchange of colours 2 and 3 are represented as blue circles and green crosses respectively.

Such states have a total momentum $K^{(\kappa)} = k_1 + k_2$ which is quantised, since $\tilde{\mathcal{T}}_L^2 = \mathbb{I}$, according to

$$K^{(\kappa)} = \frac{2\pi}{\tilde{L}}\left(n_1^{(\kappa)} + n_2^{(\kappa)}\right), \qquad \tilde{L} = 2L, \tag{A.16}$$

where $\kappa = 0$ corresponds to the odd sector and $\kappa = 1$ to the even sector. Furthermore:

$$\begin{aligned}
k_1\tilde{L} + \delta^{(\kappa)}(k_1, k_2) &= 2\pi n_1^{(\kappa)}, \\
k_2\tilde{L} + \delta^{(\kappa)}(k_2, k_1) &= 2\pi n_2^{(\kappa)},
\end{aligned} \tag{A.17}$$

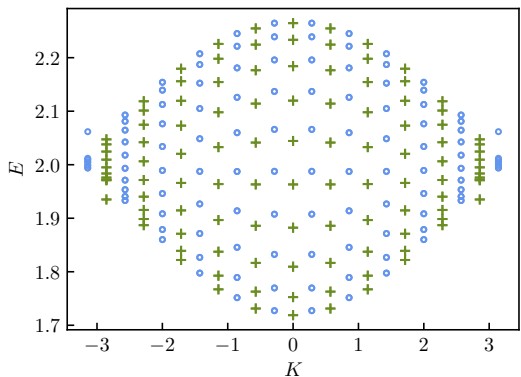

Figure 27: Two particle spectra in the *partially twisted sector* for $L = 10$ and transverse field $g = 0.2$. Green crosses indicate even states with respect to the $\mathbb{Z}_2$ symmetry under the swap $2 \leftrightarrow 3$ while blue circles indicate odd states.

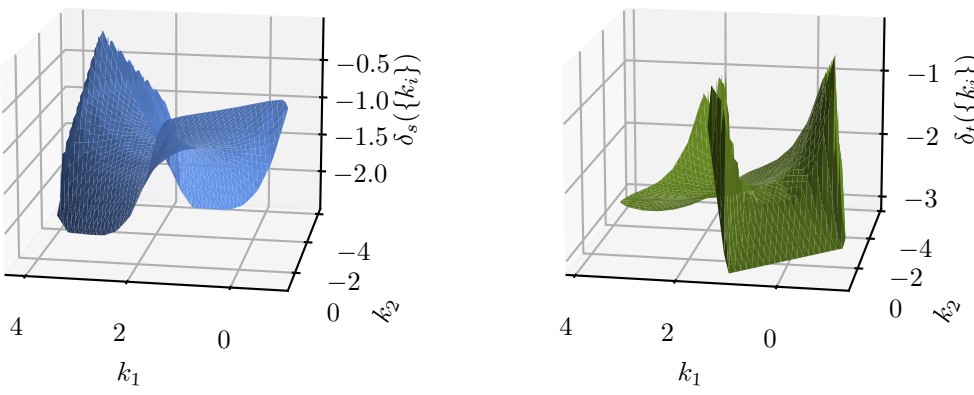

Figure 28: *Left*: Phase shift $\delta_s(k_1, k_2)$ in the odd channel (*singlet*). *Right*: Phase shift $\delta_t(k_1, k_2)$ in the even channel (*triplet*). The surfaces interpolate the solutions to (A.17) with ED data for different chain lengths ($L = 8, \ldots, 14$).

with

$$\delta^{(0)}(k_1, k_2) = \delta_s(k_1, k_2),$$
$$\delta^{(1)}(k_1, k_2) = \delta_t(k_1, k_2). \qquad (A.18)$$

We can solve (A.17) by considering the energy $E^{(\kappa)}$ of the state and finding the relative momentum $k$ by inverting

$$E = \epsilon(K/2 + k) + \epsilon(K/2 - k). \qquad (A.19)$$

Going from $(K, k)$ variables to $(k_1, k_2)$ we finally find the phase shifts $\delta^{(\kappa)}(k_1, k_2)$.

In Fig. 28 we present an interpolation of the phase shift surfaces in both sectors.

### A.3.2  Kink-kink scattering

In the partially twisted sector there exist states with non-zero topological charge. This corresponds to configurations of charged mesons and bubbles shown in Fig. 19 where we have

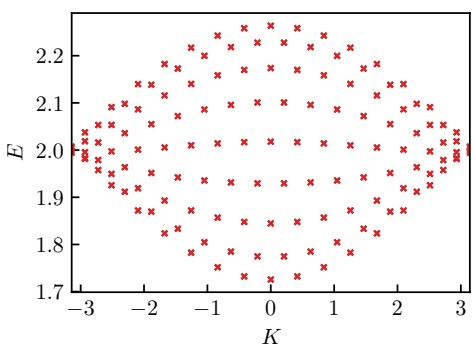
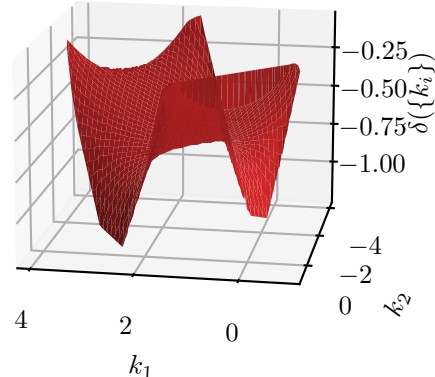

Figure 29: The energy spectrum (*left*) of a chain of length $L = 10$ and with transverse field $g = 0.2$ with twisted boundary conditions and phase-shift (*right*) $\hat{\delta}(k_1, k_2)$ for kink-kink interaction. The surface of the phase-shift interpolates the solutions to (A.23) with ED data for different chain lengths ($L = 8, \ldots, 14$).

kink-kink scattering with phase shift $\hat{\delta}(k_1, k_2)$ determined by

$$s_3(k_1, k_2) = e^{i\hat{\delta}(k_1, k_2)}, \tag{A.20}$$

introduced with (A.10).

We compute $\hat{\delta}(k_1, k_2)$ from exact diagonalisation data on a finite chain of length $L$ as in Section A.3.1. For the kink-kink channel we use twisted boundary conditions acting with the operator

$$\mathcal{T}_L = \mathbb{I}_1 \mathbb{I}_2 \cdots T_L, \tag{A.21}$$

with $T_L$ given by (7) performing the cyclic permutation $1 \to 2$, $2 \to 3$ and $3 \to 1$ on site $L$.

We select two-particle states with operator (A.6). Such states have a total momentum $K = k_1 + k_2$ which is quantised, since $\mathcal{T}_L^3 = \mathbb{I}$, according to

$$K = \frac{2\pi}{\hat{L}}(n_1 + n_2), \qquad \hat{L} = 3L. \tag{A.22}$$

Furthermore the individual momenta satisfy

$$\begin{aligned} k_1\hat{L} + \hat{\delta}(k_1, k_2) &= 2\pi n_1, \\ k_2\hat{L} + \hat{\delta}(k_2, k_1) &= 2\pi n_2, \end{aligned} \tag{A.23}$$

and we can proceed to the computation of the phase shift as in Section A.3.1. In Fig. 29 we show the ED spectrum for $L = 10$ in the left panel and the interpolated surface for the kink-kink phase shift on the right panel.

## A.4 Spontaneous magnetisation

The spontaneous magnetisation can be determined directly in the infinite volume limit using iTEBD, as explained in Appendix C. We consider the ground state polarised in the 1 direction,

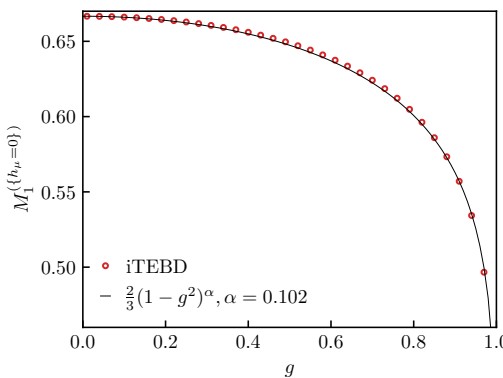

Figure 30: Spontaneous magnetisation $M_1$ for the pure transverse Potts model ($\{h_\mu = 0\}$) as a function of the coupling $g$. The red dots correspond to the iTEBD data, while the solid black line is the fitted function $\frac{2}{3}(1-g^2)^\alpha$.

for which the expectation value of $M_1$ is reported in Fig. 30 as a function of $g$, where the data are fitted with the function

$$M_1 = \frac{2}{3}(1-g^2)^\alpha \,, \tag{A.24}$$

resulting in $\alpha \approx 0.102$. While this is merely a fitting function that describes the data well in the ferromagnetic regime away from the vicinity of the critical point $g_c = 1$, the numerically obtained exponent $\alpha$ can be compared to conformal field theory. Using the fact that the temperature perturbation has scaling dimension $\Delta_\epsilon = 4/5$, while for the magnetisation operator $\Delta_\sigma = 2/15$ [64], conformal field theory predicts that the spontaneous magnetisation scales with the exponent

$$\frac{\Delta_\sigma}{2 - \Delta_\epsilon} = \frac{1}{9} = 0.111\ldots \tag{A.25}$$

for $|g - g_c| \ll 1$. This matches very well with $\alpha$, especially when taking into account that the latter was extracted using data away from $g = g_c$.

The other two magnetisations are simply given by $M_2 = M_3 = -M_1/2$.

# B Matching the semiclassical spectrum with exact diagonalisation results

Here we compare exact diagonalisation results for the spectrum of the mixed-field Potts model (2) to semiclassical results obtained in Section 3, for the parameter values applied in the quantum quenches of the main text. We demonstrate that the semiclassical approximation matches the exact diagonalisation results for a periodic chain of length $L = 10$ well, allowing for an unambiguous interpretation of the energy spectrum.

We choose the longitudinal field to align with direction 1 (*red*):

$$H(h_1, g) = -\sum_{i=1}^{L} \sum_{\mu=1}^{3} P_i^\mu P_{i+1}^\mu - \sum_{i=1}^{L} h_1 P_i^1 - g \sum_{i=1}^{L} \tilde{P}_i \,, \tag{B.1}$$

and consider both the case of positive and negative longitudinal field $h_1$. For both signs the Hamiltonian preserves the $\mathbb{Z}_2$ subgroup generated by the charge conjugation $\mathcal{C}$ exchanging colours 2 and 3, which allows for two distinct boundary conditions:

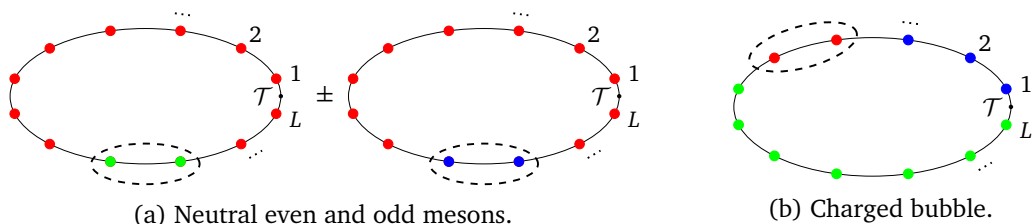

(a) Neutral even and odd mesons.          (b) Charged bubble.

Figure 31: Two-kink configurations with pTBC and a positive longitudinal field $h_1$ along the *red* direction. The twist operator $\tilde{\mathcal{T}}_L$ swaps colours *blue* and *green* at the end of the chain, but leaves *red* unchanged.

- Partially twisted boundary conditions (pTBC)
  This corresponds to inserting the twist operator $\tilde{\mathcal{T}}_L$ (A.14) swapping colours 2 and 3:

$$P^\mu_{i+L} = \tilde{\mathcal{T}}_L P^\mu_i \,, \qquad \tilde{P}_{i+L} = \tilde{\mathcal{T}}_L \tilde{P}_i \,.$$

- Periodic boundary conditions (PBC)

$$P^\mu_{i+L} = P^\mu_i \,, \qquad \tilde{P}_{i+L} = \tilde{P}_i \,.$$

All states can be classified according to their eigenvalue (parity) with respect to the unbroken $\mathbb{Z}_2$ symmetry.

## B.1 Positive longitudinal field

The possible two-kink configurations with pTBC are depicted in Fig. 31: the lowest states in energy are even and odd mesons built upon the true vacuum state, while the spectrum built over the two false vacua is instead given by charged bubbles due to the twist at the edges of the chain. The semiclassical meson spectrum is hence described by equation (51), where the energy of the meson is computed with respect to the energy of the true vacuum state, which can be computed from exact diagonalisation with periodic boundary conditions. For the bubbles, the relevant quantisation condition is given by (56), and since they are built over the false vacua, the energies given by (56) have to be shifted by the energy difference between the true vacuum and the false vacua. In finite volume, the two spectra are separated by a crossover regime, where the energy levels can be described as collisionless mesons (or bubbles) given by (43), similarly to what was found for the Ising chain [23].

In Fig. 32 we report ED data for the energy spectrum with pTBC for different values of the longitudinal field $h_1$, along with semiclassical energy levels of neutral mesons and charged bubbles. We note that the semiclassical quantisation conditions cease to have solutions for large $K > K_*$, where $K_*$ is the value when the turning point reaches the edge of the Brillouin zone. Its value depends on the quasiparticle excitation considered. To get the spectrum for all values of $K$, a full quantum mechanical description must be considered (c.f. Ref. [23] for the corresponding analysis in the Ising case).

For PBC, the relevant configurations shown in Fig. 33 include neutral even and odd mesons described by Eqs. (51), and neutral even and odd bubbles described by (53). In Fig. 34 we report ED data for the energy spectrum with PBC for two different values of the longitudinal field $h_1$, along with semiclassical energy levels of neutral mesons and bubbles.

In addition to the bound states, the two-kink sector contains further levels corresponding to kink-antikink states interpolating between the metastable vacua 2 and 3 (see Fig. 2). This subsystem can be treated as an effective Ising model, and the kinks can be considered as free

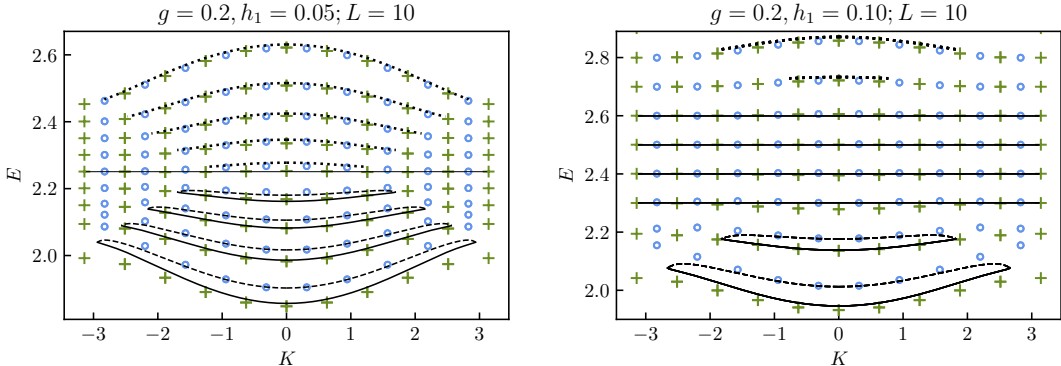

Figure 32: The semiclassical energy spectrum (solid, dashed and dotted lines) compared to exact diagonalisation results (green crosses and blue circles) with partially twisted boundary conditions (pTBC) for different positive values of the longitudinal field $h_1 = 0.05$ and $h_1 = 0.1$ and for a fixed transverse field $g = 0.2$. The length of the chain used in the exact diagonalisation procedure was chosen to be $L = 10$. The green crosses (and the blue circles) indicate $\mathcal{C}$-even (and $\mathcal{C}$-odd) states according to the unbroken $\mathbb{Z}_2$ subgroup generated by $\mathcal{C}$ in (8). The bottom part of the spectrum is described by even and odd neutral mesons (solid and dashed lines, respectively). In contrast, the top part of the spectrum is composed of charged bubbles (dotted lines). The crossover between the two regimes is given by the collisionless mesons/bubbles, which can be described either by (43) or (49). We note that the even/odd lines meet exactly when the turning point reaches $\pi$, where the dispersion relation cannot be followed semiclassically anymore.

fermions with the dispersion relation

$$\epsilon_{\text{Ising}}(k) = \sqrt{1 + h_{\text{eff}}^2 - 2h_{\text{eff}}\cos k}\,, \tag{B.2}$$

where

$$h_{\text{eff}} = \frac{2g}{3}\,, \tag{B.3}$$

with $g$ being the transverse field of the Potts model [65]. Then the free kink-antikink states of total momentum $K$ and relative momentum $k$ have total energy given by

$$E_{\text{Ising}} = \epsilon_{\text{Ising}}(K/2 + k) + \epsilon_{\text{Ising}}(K/2 - k) + \epsilon_0^{\text{FV}}\,, \tag{B.4}$$

where $k = \{\pi/L, 2\pi/L, \ldots, \pi\}$, and $\epsilon_0^{\text{FV}}$ is the energy of the false vacuum states relative to the true vacuum. As shown in Fig. 34, the spectrum (B.4) (depicted by dashed grey lines) matches the appropriate energy levels. In the thermodynamic limit, these states give rise to a two-particle continuum. As explained in the main text in Subsection 4.1.3, the collisional bubbles lying in the energy range of the effective Ising two-kink levels hybridise with the effective Ising two-kink states and form a continuum.

## B.2 Negative longitudinal field

Reversing the sign of the longitudinal field $h_1$, the two-kink configurations allowed for twisted boundary conditions are changed. Now the lowest energy states are charged mesons over the degenerate true vacua described by Eq. (55), while from the unique false vacuum, odd and even bubbles described by Eq. (53) can form. These configurations are illustrated in Fig. 35.

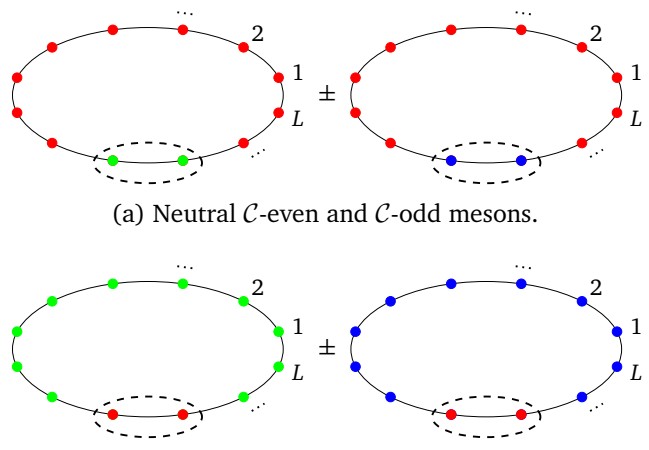

(a) Neutral $\mathcal{C}$-even and $\mathcal{C}$-odd mesons.

(b) Neutral $\mathcal{C}$-even and $\mathcal{C}$-odd bubbles.

Figure 33: Neutral meson/bubble configurations with PBC and a positive longitudinal field $h_1$ along the *red* direction.

As in the previous case, the crossover regime is given by collisionless mesons (or bubbles) described by Eq. (43). In Fig. 36 we report ED data for the energy spectrum together with the semiclassical energy levels of charged mesons and neutral bubbles.

When considering periodic boundary conditions, the semiclassical spectrum is identical to the $h_1 > 0$ case, with the allowed configurations depicted in Fig. 37, corresponding to neutral mesons (51) and neutral bubbles (53) shown by dashed black lines.

The corresponding ED data for the energy spectrum are reported in Fig. 38, together with the semiclassical energy levels of neutral mesons and neutral bubbles.

Furthermore, there are now effective Ising kink-antikink states corresponding to the two degenerate true vacua, with their energy levels for total momentum $K$ given by

$$E_{\text{Ising}} = \epsilon_{\text{Ising}}(K/2 + k) + \epsilon_{\text{Ising}}(K/2 - k), \tag{B.5}$$

where $k = \{\pi/L, 2\pi/L, \ldots, \pi\}$. The corresponding energy levels are shown by dashed grey lines in Fig. 33. Again, these states form a two-particle continuum in the thermodynamic limit. Similarly to the case of the positive longitudinal field, the collisional mesons lying in the energy range of the effective Ising two-kink levels hybridise with the effective Ising two-kink states and form a continuum.

## C Details of the iTEBD calculations

For the computation of the spontaneous magnetisation in Fig. 30 an imaginary time evolution with the infinite Time Evolving Block Decimation (iTEBD) algorithm was performed. The system is initially prepared in the state

$$|\psi_0\rangle = \bigotimes_{i=1}^{L} |1\rangle_i . \tag{C.1}$$

We then apply the non unitary operator

$$U(T) = e^{-H(0,g)T} |\psi_0\rangle , \tag{C.2}$$

for different values of $g$, with $H$ being the pure transverse Potts Hamiltonian (A.1). After a sufficiently long period of time $T$, the system relaxes to its ground state polarised in direction 1 [66], and the expectation value of the operator $P^1$ can be computed.

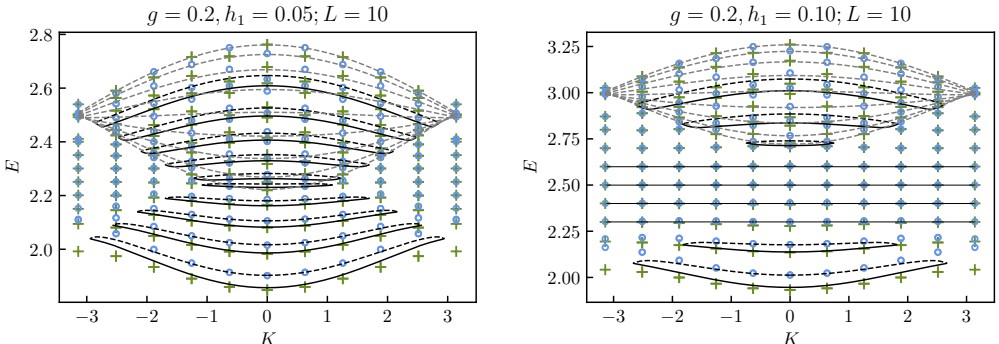

Figure 34: The semiclassical energy spectrum (solid, dashed and dotted lines) compared to exact diagonalisation results (green crosses and blue circles) with periodic boundary conditions (PBC) for different positive values of the longitudinal field $h_1 = 0.05$ and $h_1 = 0.1$ and for a fixed transverse field $g = 0.2$. The length of the chain used in the exact diagonalisation procedure was chosen to be $L = 10$. The green crosses (and the blue circles) indicate $\mathcal{C}$-even (and $\mathcal{C}$-odd) states according to the unbroken $\mathbb{Z}_2$ subgroup generated by $\mathcal{C}$ of (8). The bottom part of the spectrum is described by odd and even neutral mesons (solid and dashed black lines respectively) of (51) while the top part of the spectrum is described by even and odd neutral bubbles (solid and dashed black lines respectively) given by (53). The crossover of the two regimes is described by the collisionless mesons (or bubbles) of (43) or (49). Additionally, the grey lines on the top part of the spectrum correspond to the two-kink states above the two degenerate false vacua described by effective Ising kink-antikink states as in (B.4).

The imaginary time evolution is performed by a second-order Trotter-Suzuki approximation with Trotter steps $\delta t = 0.01$, while the total time of evolution $T$ is chosen by computing the energy eigenvalue $\varepsilon(t)$ of the state at each time step $t$ and the time evolution stops when the difference between two subsequent energy measurements is less than $10^{-14}$, that is:

$$|\varepsilon(T) - \varepsilon(T - \delta t)| < 10^{-14}. \tag{C.3}$$

The maximum bond dimension was set to $\chi_{\max} = 81$, a value never reached by the actual bond dimension used in the simulations.

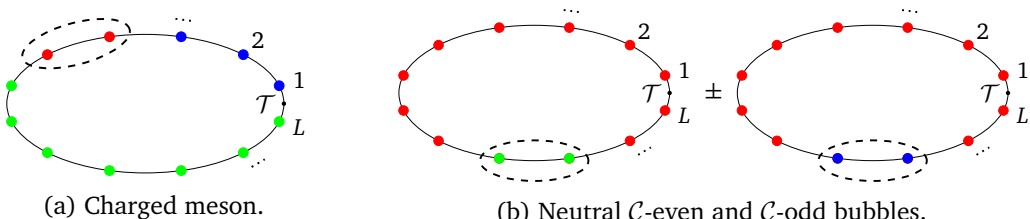

(a) Charged meson.  (b) Neutral $\mathcal{C}$-even and $\mathcal{C}$-odd bubbles.

Figure 35: Two-kink configurations with pTBC and a negative longitudinal field $h_1$ along the *red* direction.

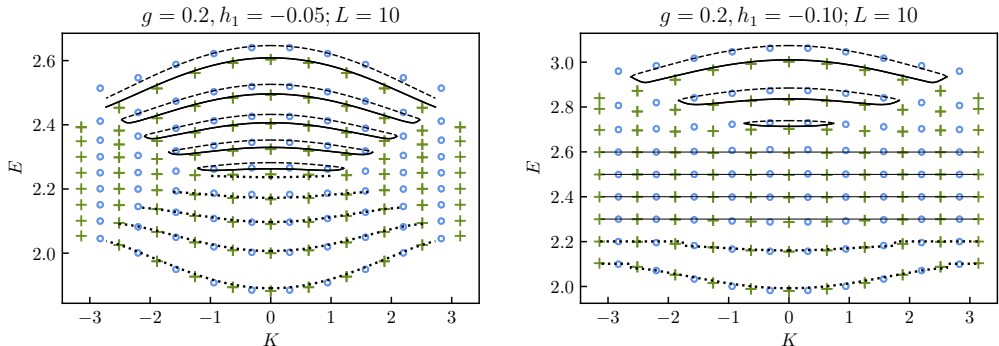

Figure 36: The semiclassical energy spectrum (solid, dashed and dotted lines) compared to exact diagonalisation results (green crosses and blue circles) with partially twisted boundary (pTBC) conditions for different negative values of the longitudinal field $h_1 = -0.05$ and $h_1 = -0.1$ and for a fixed transverse field $g = 0.2$. The length of the chain used in the exact diagonalisation procedure was chosen to be $L = 10$. The green crosses (and the blue circles) indicate $\mathcal{C}$-even (and $\mathcal{C}$-odd) states according to the unbroken $\mathbb{Z}_2$ subgroup generated by $\mathcal{C}$ of (8). The bottom part of the spectrum is described by charged mesons of (55) (dotted lines) while the top part of the spectrum is described by even and odd neutral bubbles (solid and dashed lines respectively) given by (53). The crossover between the two regimes is given by the collisionless mesons (or bubbles) of (43) or (49).

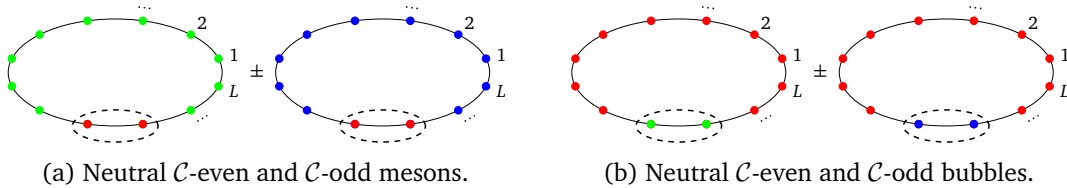

(a) Neutral $\mathcal{C}$-even and $\mathcal{C}$-odd mesons.  (b) Neutral $\mathcal{C}$-even and $\mathcal{C}$-odd bubbles.

Figure 37: Neutral meson/bubble configurations with PBC and a negative longitudinal field $h_1$ along the *red* direction.

To perform the quantum quenches and obtain the numerical results discussed in Section 2.2, 2.3 and 4.1, we used again the iTEBD method, this time for real-time evolution with second-order Trotterisation. As previously discussed, the initial state was always chosen as

$$|\psi_0\rangle = \bigotimes_{i=1}^{L} |1\rangle_i \, , \tag{C.4}$$

and we time evolved this state by applying the corresponding Hamiltonian (2) with a time step $\delta t = 0.005$. We performed simulations with $\chi_{\max} = 800$ up to shorter times (approximately $t \approx 200$) and with $\chi_{\max} = 300$ up to longer times ($t = 1000$). The larger maximum bond dimension ensures higher precision, but it significantly reduces the speed of the simulations, preventing long-time runs. However, longer-time simulations prove essential as they show that even for the standard confining/anticonfining quenches, the entanglement entropy $S$ does not saturate completely, displaying a slow drift instead. In contrast, for partial confining and anticonfining quenches the entanglement entropy does not saturate at all. For the four different quench types, $S$ is shown in Figs. 39, 40 where simulation results with bond dimension $\chi_{\max} = 800$ are shown in blue, while those with $\chi_{\max} = 300$ are shown in orange.

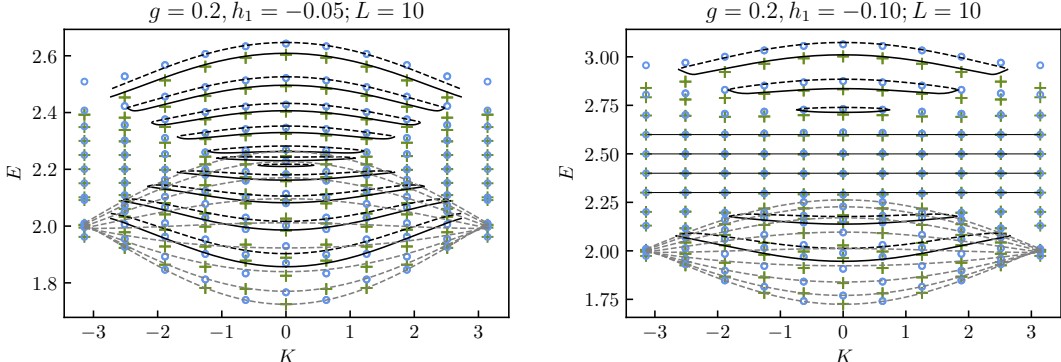

Figure 38: The semiclassical energy spectrum (solid, dashed and dotted lines) compared to exact diagonalisation results (green crosses and blue circles) with periodic boundary conditions (PBC) for different negative values of the longitudinal field $h_1 = -0.05$ and $h_1 = -0.1$ and for a fixed transverse field $g = 0.2$. The length of the chain used in the exact diagonalisation procedure was chosen to be $L = 10$. The green crosses (and the blue circles) indicate $\mathcal{C}$-even (and $\mathcal{C}$-odd) states according to the unbroken $\mathbb{Z}_2$ subgroup generated by $\mathcal{C}$ of (8). The bottom part of the spectrum is described by odd and even neutral mesons (solid and dashed black lines respectively) of (51) while the top part of the spectrum is described by even and odd neutral bubbles (solid and dashed black lines respectively) given by (53). The crossover of the two regimes is described by the collisionless mesons (or bubbles) of (43) or (49). Additionally, the grey lines on the bottom part of the spectrum are in correspondence with the two-kink states above the two degenerate true vacua that constitute non-localised states and are described by effective Ising kink-antikink states of (B.5).

Additionally, long-time simulations are needed to obtain sufficient frequency resolution for quench spectroscopy. Here, we compare the results obtained with the choice of $\chi_{\max} = 300$ to those with larger bond dimension $\chi_{\max} = 800$ to verify their agreement. In the initial time period for which both results are available, the difference between the time evolution of the magnetisation and the entropy is negligible, and the Fourier spectra qualitatively show the same features. We present an example of the Fourier spectrum of $M_1(t)$ for each of our four quench scenarios performed with bond dimension $\chi_{\max} = 800$ in Figs. 41-44, shown side-by-side with the $\chi_{\max} = 300$ results for easier comparison. The data demonstrate that the two sets of results are consistent, with the longer simulation time increasing frequency resolution, which is also essential for detecting the presence of baryonic excitations. In the main text, we only include the results of the simulations with $\chi_{\max} = 300$.

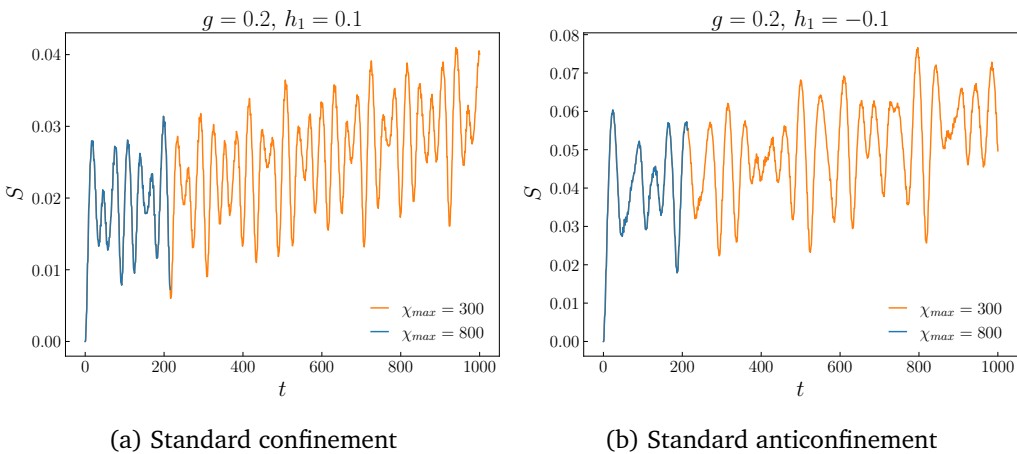

(a) Standard confinement        (b) Standard anticonfinement

Figure 39: Long-term time evolution of the entanglement entropy in the standard confining case for the parameters $g = 0.2$ and $h_1 = 0.1$ (left) and in the standard anticonfining case for $g = 0.2$ and $h_1 = -0.1$ (right). The orange curves denote the results of the simulations with the choice of $\chi_{\max} = 300$, and the blue curves the results with $\chi_{\max} = 800$. In both cases, the time evolution of the entropy exhibits a slow drift on long time scales.

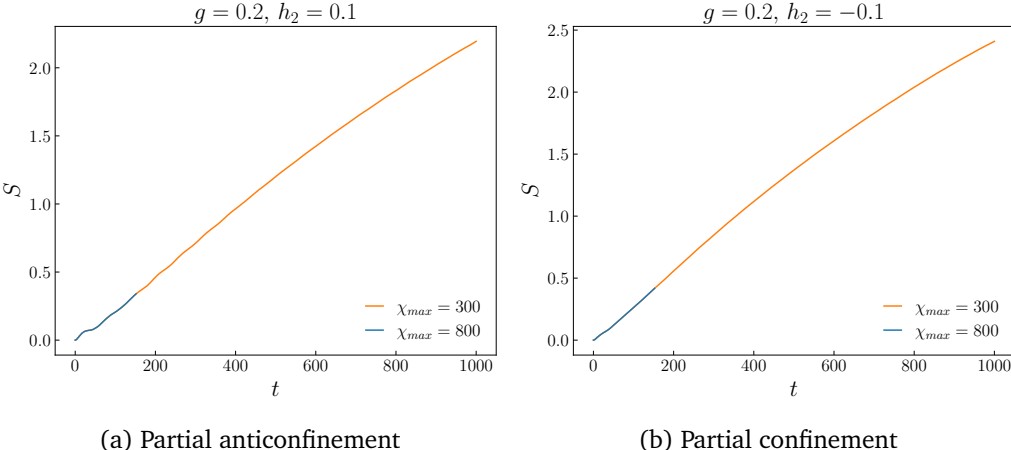

(a) Partial anticonfinement        (b) Partial confinement

Figure 40: Time evolution of the entanglement entropy in the partial anticonfining case (positive oblique) for the parameters $g = 0.2$ and $h_2 = 0.1$ (left) and in the partial confining case (negative oblique) for $g = 0.2$ and $h_2 = -0.1$ (right). The orange curves denote the results of the simulations with the choice of $\chi_{\max} = 300$, and the blue curves the results with $\chi_{\max} = 800$. In both cases, the time evolution of the entropy shows unsuppressed growth on long time scales.

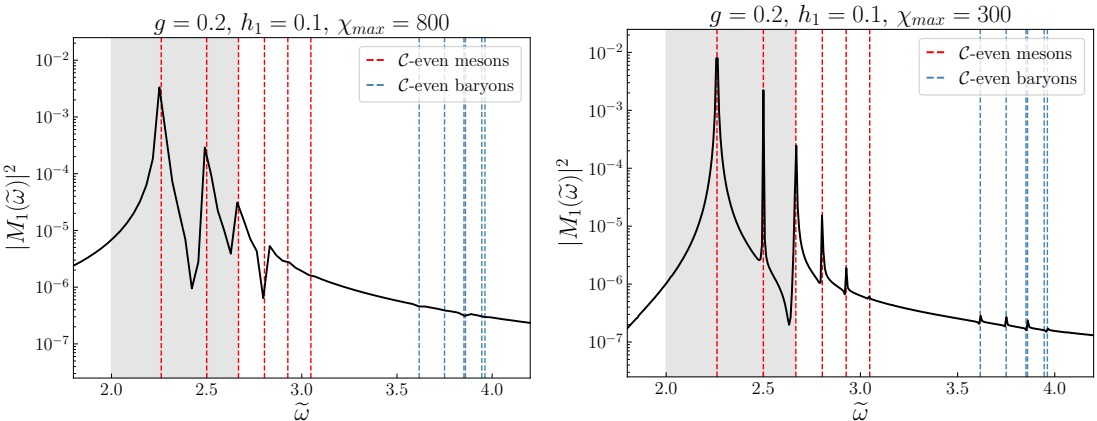

Figure 41: The Fourier spectrum of $M_1$ in terms of $\widetilde{\omega} = \omega/m_k$ in the standard confining case for the parameters $g = 0.2$, $h_1 = 0.1$, $\chi_{\max} = 800$ (left) and $\chi_{\max} = 300$ (right). The dashed red lines correspond to the respective low energy $\mathcal{C}$-even meson masses, while the dashed blue lines denote the low energy $\mathcal{C}$-even baryon masses. The masses were calculated via ED with $L = 10$ and PBC as shown in Fig. 34, and they match very well with the position of the peaks. The background is coloured grey in the interval $(\tilde{\omega}_{\min}, \tilde{\omega}_{\max}) = (2, 2.663)$; collisional/collisionless mesons lie inside/outside this interval, respectively.

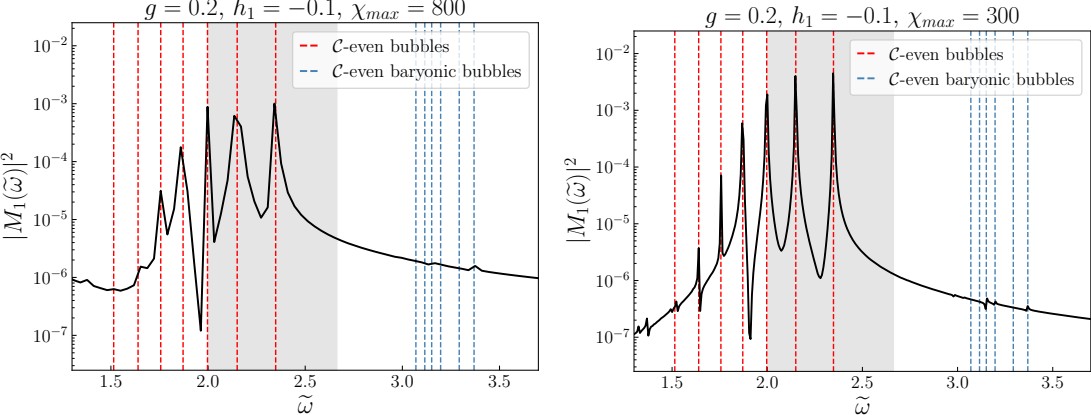

Figure 42: The Fourier spectrum of $M_1$ in terms of $\widetilde{\omega} = \omega/m_k$ in the standard anticonfining case for the parameters $g = 0.2$, $h_1 = -0.1$, $\chi_{\max} = 800$ (left) and $\chi_{\max} = 300$ (right). The dashed red lines correspond to the respective first few $\mathcal{C}$-even bubble masses, while the dashed blue lines denote the first few $\mathcal{C}$-even baryonic bubble masses. The masses were calculated via ED with PBC and with $L = 10$ as shown in Fig. 38, and the energy of the false vacuum is subtracted from the ED masses. The ED masses match very well with the position of the peaks. The background is coloured grey in the interval $(\tilde{\omega}_{\min}, \tilde{\omega}_{\max}) = (2, 2.663)$; collisional/collisionless bubbles lie inside/outside this interval, respectively.

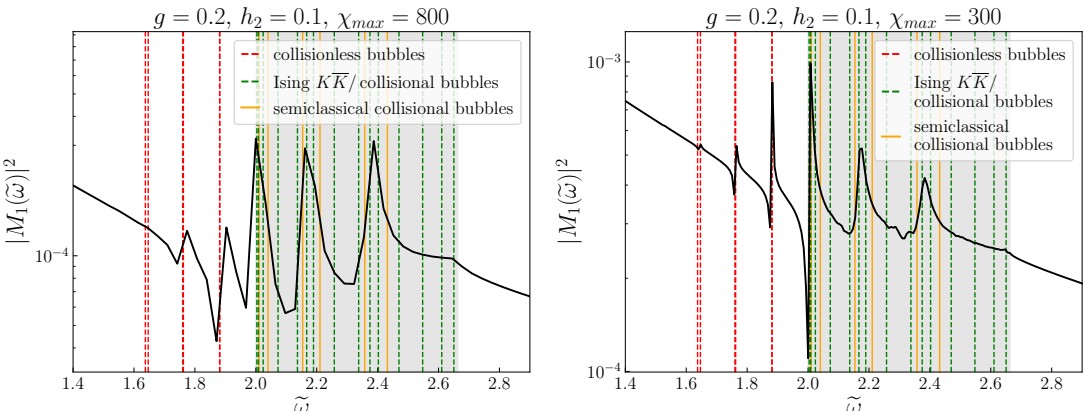

Figure 43: The Fourier spectrum of $M_1$ in terms of $\widetilde{\omega} = \omega/m_k$ in the partial anticonfining case for the parameters $g = 0.2$, $h_2 = 0.1$, $\chi_{\max} = 800$ (left) and $\chi_{\max} = 300$ (right). The dashed lines denote the high-energy (both $\mathcal{C}$-even and $\mathcal{C}$-odd) 2-kink masses. The green lines correspond to the hybridised Ising $K\bar{K}$ and collisional bubble states, while the red ones to the collisionless bubbles. The red lines match very well with the sharp peaks, while the top edge of the two-kink continuum (the highest energy green line) corresponds to a small cusp in the quench spectrum. The positions of the wide resonant peaks are indeed in the vicinity of the semiclassical collisional bubble masses denoted by solid orange lines. The background is coloured grey in the interval $(\tilde{\omega}_{\min}, \tilde{\omega}_{\max}) = (2, 2.663)$ where the collisional bubbles lie. All masses were calculated via ED with PBC and $L = 10$ relative to the energy of the false vacuum as shown in Fig. 34.

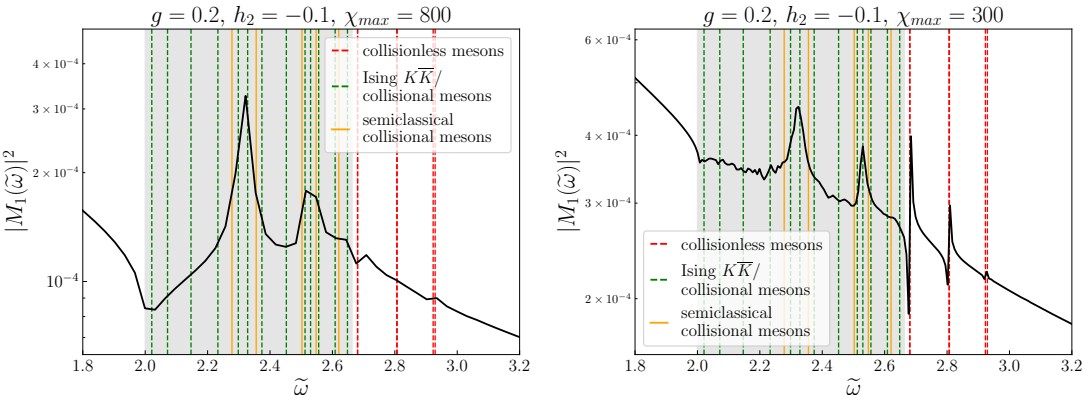

Figure 44: The Fourier spectrum of $M_1$ in terms of $\widetilde{\omega} = \omega/m_k$ in the partial confining case for the parameters $g = 0.2$, $h_2 = -0.1$, $\chi_{\max} = 800$ (left) and $\chi_{\max} = 300$ (right). The dashed lines denote the low-energy (both $\mathcal{C}$-even and $\mathcal{C}$-odd) 2-kink masses. The green lines correspond to the hybridised Ising $K\bar{K}$ and collisional meson levels, while the red ones show the collisionless mesons. The red lines match very well with the sharp peaks, while the bottom edge of the two-kink continuum (the lowest energy green line) corresponds to a small cusp in the quench spectrum. The positions of the wide resonant peaks are indeed in the vicinity of the semiclassical collisional meson masses denoted by solid orange lines. The background is coloured grey in the interval $(\tilde{\omega}_{\min}, \tilde{\omega}_{\max}) = (2, 2.663)$ where the collisional mesons lie. All masses were calculated via ED with PBC and with $L = 10$ as shown in Fig. 38.

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
