# Peer review of "Confinement and false vacuum decay on the Potts quantum spin chain"

_SciPost Physics, doi:SciPost Phys. 18, 082 (2025)_

## Round 1 · Referee Report · Sergei Rutkevich (Referee 1) · 2024-12-19

Report
The manuscript meets the SciPost Physics acceptance criteria.
I recommend its publication in SciPost Physics with minor revision in response to my two comments.
Requested changes
1- I propose to add vertical gridlines in Figures 5.1 -5.4 indicating the boundaries of the continuous two-kink spectrum in the infinite unperturbed chain.
2 - I think, that the interpretation of the computer simulations in the oblique regimes should be modified, as it is explained in the report.
Recommendation
Publish (surpasses expectations and criteria for this Journal; among top 10%)

---

## Round 1 · Referee Report · Anonymous (Referee 2) · 2025-1-4

Report
In this work the authors study the non-equilibrium dynamics of the three-state Potts quantum chain in the presence of mixed fields. As in the previously studied quantum Ising chain, a rich physical phenomenology including confinement is expected, and indeed observed, in this case. Furthermore, the 3-state Potts chain offers a wider scope of quench protocols, which result in a richer physics (which, borrowing from the QCD terminology, are associated with "baryonic excitations").
Starting from a purely ferromagnetic initial state and with an external magnetic field pointing in one of the three directions there are four possible quench protocols (depending on the relative orientation of the initial magnetization with the external longitudinal magnetic field, and of the sign of the latter), each of which comes with a different phenomenology (light-cone propagation, standard or partial confinement or anticonfinement). Numerical simulations by iTEBD are presented for each case, and the numerical data for the entanglement entropy, average magnetizations or correlation functions, are confronted with a semi-classical analysis.
Importantly, the model is not integrable on the lattice, therefore requiring a numerical evaluation of the various scattering phases. Combined with a semi-classical analysis, this results in a very precise estimation of (a significant part of) the low-lying energy spectrum in the various regimes, which is used to provide a compelling interpretation of the non-equilibrium spectral data obtained from numerics.
Altogether this is a very complete study of a new and interesting problem, completing the state-of the art (the case of the Ising model) with a richer phenomenology (even though the analogy to QCD may be somewhat artificial).
I recommend its publication in SciPost physics once the comments below have been considered.
Requested changes
1-Eq. (2.2) : in the definition of the Hamiltonian, I suggest that the authors recall which term break integrability, and how this compares with the Ising case.
2-A significant part of the interesting physics (relying on Bloch oscillations and Wannier-Stark localisation) goes as in the previously studied case of the Ising model, and therefore very little background is given. I suggest to add a few lines of explanation of the Wannier-Stark localisation phenomenology, and why it is expected to hold in the present model, as opposed to other quantum spin chains.
3- Appendix A.4 : since the underlying CFT is known, I find surprising that no exact estimation of $\alpha$ can be found. Could the authors comment on that ?
4- Fig A.3: Could the authors confront their phase shift with those known from field theory ?
5- Eq (4.37) and following : notation $\mathcal{E}_n$ for the energy is unexplained. Is it the same as $E_n$ ? I so notations should be harmonized
Recommendation
Ask for minor revision

---

## Round 1 · Referee Report · Anonymous (Referee 3) · 2025-1-9

Report
The authors study non-equilibrium physics in the mixed-filed 3-state Potts model in 1+1D. The authors observe rich dynamics compare to the Ising model. In particular, the authors claim that in addition to Wannier-Start localization and dynamical confinement, they observe baryonic excitations.
I find the subject interesting and timely, though the explanations are in many places are too lengthy and full of technical details making it hard to follow the general idea. For instance, the definition of the Potts model (very standard and well known in the field) takes two full pages. And then it is repeated again but with the longitudinal field in Eq.(3.3), and again in Eq.(3.5), and again in Eq.(4.47)
I'd recommend to significantly shorten the main part, perhaps, moving technical details into appendices (obviously, this makes the paper very pedagogical and contributes to the reproducibility of the results, but the way it is written now, the amount of technical information included distracts the attention from the main results)
I do not see any conceptual difference between model of Eq. (3.3)/sketches 3.1 and the model in Eq. (3.5)/sketches 3.2. By applying an overall rotation on one model, one man make the two of the identical. Do I miss something?
The captions are minimalistic. I strongly recommend the authors to include more details in the description of their results and to guide readers on what they could (or could not) see in the Figures.
I appreciate the details on TEBD in the appendix, still minimal information on the used truncation error/bond dimension can be mentioned in the beginning of the section 3.2.
In fig.4.8, 4.10, I guess, symbols mean exact diagonalization results and lines are from semiclasics? This should be mentioned.
Fourrier spectrum is not properly defined in the main text (one Eq. In the main text and all technical details in the appendix would be great, I think)
In Fig.5.5/5.6 the correlations are non-zero even outside the light-cone. The reason for this should be briefly mentioned in the text.
I find the last sentence “Finally, it would be interesting to see whether the
various phenomena explored here can be realized in experimental settings, e.g., by ultra-cold atomic
simulators.” too vague. The authors should either remove it or support it with some concrete ideas.
I do recommend the publication of the present study in SciPostPhysics once the authors have considered the comments above.
Recommendation
Publish (meets expectations and criteria for this Journal)

---

## Round 2 · Referee Report · Sergei Rutkevich (Referee 1) · 2025-2-9

Report
I recommend publication of this paper in SciPost Physics.
Recommendation
Publish (surpasses expectations and criteria for this Journal; among top 10%)

---

## Round 2 · Author Response

Herein, we resubmit our manuscript entitled "Confinement and false vacuum decay on the Potts quantum spin chain".
We are grateful to the Referees for their thorough reading of the manuscript and valuable comments, which helped us improve our presentation. Our detailed reply is given in the list of changes.
Yours sincerely,
Gábor Takács

---

## Round 2 · List of Changes

We present our replies below; we also made several small changes (correcting typos and grammar), which we do not list separately.
Replies to Report #1
- I propose to add vertical gridlines in Figures 5.1 -5.4, indicating the boundaries of the continuous two-kink spectrum in the infinite unperturbed chain.
Response: We modified the requested figures by colouring the background grey in the interval (ω̃min , ω̃max ). Because of the change in the structure of the paper, the new numbers of the figures are Fig. 4.1-4.4. We also explain the modification in the main text in Subsection 4.1.1 and note this in the captions of the relevant figures. Furthermore, we modified Figs. C.3-C.6 accordingly.
Change: We modified Figs. 4.1-4.4 and Figs. C.3-C.6 by colouring the background grey in the interval (ω̃min , ω̃max ) and we also added a sentence to the captions of the figures to note this. We explain why we highlight this interval in the main text in Subsection 4.1.1.
- I think that the interpretation of the computer simulations in the oblique regimes should be modified, as explained in the report.
Response: We agree with the Referee’s interpretation of the spectrum that the collisional bubbles/mesons hybridise with the effective Ising two-kink states and modify the text accordingly. We explained this idea in Subsection 4.1.3 and also mentioned it briefly in Subsections 4.1.4 and, regarding the explanation of the ED spectrum, at the end of Subsection B.1 and B.2. We now show all the masses corresponding to hybridised effective Ising two-kink and collisional bubble/meson states by green dashed lines in Figs. 4.3-4.4 and in Figs. C.5-C.6. We also agree that the unstable (collisional) mesons/bubbles should manifest themselves as wide resonant peaks that we see in the Fourier spectra. We expect these peaks to be close to the semiclassically calculated collisional meson/bubble masses (which is indeed the case); therefore, we plotted these in Figs. 4.3-4.4 and in Figs. C.5-C.6 by vertical orange solid lines. We remark that changing the system size in iTEBD is not an option since the method works in infinite volume. Resolution of the Fourier transform can be improved only by extending simulation time, which is limited by entanglement growth and Trotter errors.
Change: Added brief explanation on the hybridised states in Subsection 4.1.3 and briefly mentioned it in Subsections 4.1.4, B.1 and B.2. Changed the colours of the dashed lines denoting the hybridised ED states to green in Figs. 4.3-4.4 and in Figs. C.5-C.6. In the same figures, we plotted the semiclassically calculated collisional meson/bubble masses by orange solid lines.
- The Referee also suggested adding four references.
Response: Regarding the first reference, the oscillations it considers are for non-confining models and so it does not seem to be relevant in our context. However, we agree that references 2-4 are relevant to our work, and we are grateful to the Referee for pointing them out.
Change: We added references 2-4 to Subsections 3.1.1 and 3.2.1 appropriately.
Replies to Report #2
- Eq. (2.2): in the definition of the Hamiltonian, I suggest that the authors recall which term breaks integrability, and how this compares with the Ising case.
Response: No specific term breaks integrability since the q = 3 Potts quantum chain is integrable only in the critical point. This statement was made clearer in the text when the model is introduced after Eq. (2.2).
Change: We added an explanatory sentence to the text at the end of the paragraph following Eq. (2.2).
- A significant part of the interesting physics (relying on Bloch oscillations and Wannier-Stark localisation) goes as in the previously studied case of the Ising model, and therefore very little background is given. I suggest to add a few lines of explanation of the Wannier- Stark localisation phenomenology and why it is expected to hold in the present model, as opposed to other quantum spin chains.
Response: A sentence was added in the introduction, briefly explaining the phenomenology.
Change: We added text in the 2nd paragraph of the introduction.
- Appendix A.4: since the underlying CFT is known, I find it surprising that no exact estimation of α can be found. Could the authors comment on that?
Response: Since the fit goes over all the range 0 < g < 1, it is well beyond the vicinity of the critical point and, therefore, need not be close to the critical exponent. In fact, we do not have data very close to g = 1 due to numerical difficulties. Additionally, the dependence on $(1−g)^2$ instead of the simpler $1−g$ is just an educated guess, making the fit much better. Nevertheless, the extracted exponent is indeed close to the CFT prediction 1/9.
Change: We added a comparison to the CFT prediction in Appendix A.4.
- Fig A.3: Could the authors confront their phase shift with those known from field theory?
Response: This was performed previously in one of our references (A. Rapp, P. Schmitteckert, G. Takacs, and G. Zarand, New J. Phys. 15 (2013) 013058), with the result that the lattice phase shifts show some significant differences from the field theory ones. This was later explained by contributions from irrelevant operators. Due to these earlier results, we already knew that this comparison would not be of much interest, so we have forgone it.
Change: We added further text to clarify the situation in the first paragraph of Subsection 3.3.
- Eq (4.37) and following: notation \mathcal{E}_n for the energy is unexplained. Is it the same as E_n? If so, notations should be harmonized.
Response: The different notation was used to distinguish between mesonic excitations, built upon the true vacuum, and bubble excitations, built from the false vacuum. They coexist in the same spectrum and have different values, so a different symbol is needed. A sentence has been added to the text when the bubble notation is introduced to make the distinction between them more explicit.
Change: We added the explanatory text “The notation $\mathcal{E}_n $, as opposed to $E_n$, is used to distinguish bubble excitations from earlier mesonic ones.”
Replies to Report #3
- For instance, the definition of the Potts model (very standard and well known in the field) takes two full pages. And then it is repeated again but with the longitudinal field in Eq.(3.3), and again in Eq.(3.5), and again in Eq.(4.47).
Response: In our view, the detailed introduction of the Potts model helps the reader understand the paper; therefore, we find it more beneficial to keep it but change it into a subsection instead of a separate section. After restructuring the paper, this is Subsection 2.1. We also removed the unnecessary repetitions of the Hamilton operator from the section discussing the quench protocols. We kept only one of the listed formulas, originally (4.47), to ensure greater clarity and connection, and moved this formula to the appendix B.1. Moreover, we do agree with the Referee's suggestion to focus more on the new scientific results, so we restructured the paper as discussed in the next response.
Change: Removed formulas (3.3) and (3.5), restructured the text, and moved (4.47) to appendix B.1.
- I’d recommend to significantly shorten the main part, perhaps, moving technical details into appendices (obviously, this makes the paper very pedagogical and contributes to the reproducibility of the results, but the way it is written now, the amount of technical infor- mation included distracts the attention from the main results).
Response: We agree with the Referee’s suggestion, and have taken steps to streamline the presentation.
Change: We moved the comparison between ED and semiclassics (originally Section 4.5) to Appendix B. On the other hand, we kept the semiclassical approximation in the main text as it contains new results relevant to the flow of the argument.
- I do not see any conceptual difference between model of Eq. (3.3)/sketches 3.1 and the model in Eq. (3.5)/sketches 3.2. By applying an overall rotation on one model, one can make the two of them identical. Do I miss something?
Response: Although the post-quench Hamiltonian is the same by applying an overall rotation, the initial state is still the pure ferromagnetic state along one fixed direction. This is what ultimately results in the difference between aligned and oblique quenches. To make this point more precise, we added a reminder of the initial state to the vacuum configuration sketches of Fig. 3.1 and Fig. 3.2 and in their captions. We also added a sentence at the end of Subsection 2.2 to emphasize the main difference between the quench scenarios once again.
Change: Addition of reminder of the initial state in the quench protocol in Figures 3.1 and 3.2 and relative captions. Addition of a clarifying sentence at the end of Subsection 2.2.
- The captions are minimalistic. I strongly recommend the authors to include more details in the description of their results and to guide readers on what they could (or could not) see in the Figures.
Response: We supplemented the captions of the figures in Subsection 2.3 and in Appendix B depicting the time evolution of the magnetisations and entanglement entropy and also the caption of Fig. 2.5 with brief comments on the behaviour of the time evolution. We also added a sentence to the captions of all of the figures showing Fourier spectra to note that the locations of the peaks match very well with the relevant ED masses.
Change: Changed the caption of Figs. 2.4-2.6, 2.8, 2.10, 2.12, C.1., C.2. to include brief information about the behaviour of the respective time evolutions. Added a sentence to the caption of Figs. 4.1-4.4, C.3-C.6 to note that the locations of the peaks in the Fourier spectra match very well with the relevant ED masses.
- I appreciate the details on TEBD in the appendix, still minimal information on the used truncation error/bond dimension can be mentioned in the beginning of the section 3.2.
Response: We included a sentence at the beginning of Subsections 2.3 and 4.1 to specify the order of Trotterisation, maximal bond dimension χmax and time step δt applied in the iTEBD simulations discussed in the subsections.
Change: One sentence was added at the beginning of both Subsections 2.3 and 4.1 about the iTEBD parameters.
- In fig.4.8, 4.10, I guess, symbols mean exact diagonalization results and lines are from semiclassics? This should be mentioned.
Response: Captions of Figures originally 4.8, 4.10, 4.12 and 4.14 have been improved accordingly. The modified numbers of the figures after restructuring are Figs. B.2, B.4, B.6 and B.8.
Change: “The semi-classical energy spectrum compared to exact diagonalisation results” changed to “The semi-classical energy spectrum (solid, dashed and dotted lines) compared to exact diagonalisation results (green crosses and blue circles)”.
- The Fourier spectrum is not properly defined in the main text (one Eq. In the main text and all technical details in the appendix would be great, I think)
Response: We agree and moved the definition of the Fourier transform forward.
Change: The definition of the Fourier transform was moved to Subsection 4.1 from the Appendix.
- In Fig.5.5/5.6, the correlations are non-zero even outside the light cone. The reason for this should be briefly mentioned in the text.
Response: The strict light cone drawn in the figure starts from a single point, while the initial correlations do have a non-zero, albeit short range.
Change: We added a short explanatory text in 4.2.1.
- I find the last sentence, “Finally, it would be interesting to see whether the various phenomena explored here can be realized in experimental settings, e.g., by ultra-cold atomic simulators.” too vague. The authors should either remove it or support it with some concrete ideas.
Change: We added a few references and reformulated the sentence to make it more concrete.

---

## Editorial Decision

published